# CAR-T cell therapy targeting surface expression of TYRP1 to treat cutaneous and rare melanoma subtypes

Sameeha Jilani[1,11], Justin D. Saco[1,11], Edurne Mugarza[1,11], Aleida Pujol-Morcillo[1], Jeffrey Chokry[1], Clement Ng[1], Gabriel Abril-Rodriguez[1,2], David Berger-Manerio[1], Ami Pant[3], Jane Hu[3], Rubi Gupta[1], Agustin Vega-Crespo[1], Ignacio Baselga-Carretero[1], Jia M. Chen [1,2], Daniel Sanghoon Shin[4,5,6], Philip Scumpia [7,8], Roxana A. Radu [3], Yvonne Chen[2,6,9,10], Antoni Ribas [1,2,6,10] & Cristina Puig-Saus [1,2,6,10] ✉

A major limitation to developing chimeric antigen receptor (CAR)-T cell therapies for solid tumors is identifying surface proteins highly expressed in tumors but not in normal tissues. Here, we identify Tyrosinase Related Protein 1 (TYRP1) as a CAR-T cell therapy target to treat patients with cutaneous and rare melanoma subtypes unresponsive to immune checkpoint blockade. TYRP1 is primarily located intracellularly in the melanosomes, with a small fraction being trafficked to the cell surface via vesicular transport. We develop a highly sensitive CAR-T cell therapy that detects surface TYRP1 in tumor cells with high TYRP1 overexpression and presents antitumor activity in vitro and in vivo in murine and patient-derived cutaneous, acral and uveal melanoma models. Furthermore, no systemic or off-tumor severe toxicities are observed in an immunocompetent murine model. The efficacy and safety profile of the TYRP1 CAR-T cell therapy supports the ongoing preparation of a phase I clinical trial.

Immune checkpoint blockade (ICB) therapy has changed the treatment paradigm for melanoma. Frontline therapy for patients with melanoma includes the combination of Cytotoxic T-Lymphocyte Antigen 4 (CTLA-4) and Programmed Cell Death Protein-1 (PD-1) blockade, which show objective response rates of 57.6% in patients with cutaneous melanoma[1], a progression-free survival of 11.5 months, and an overall survival rate of 52% at the 5-year follow-up[2,3]. Despite these remarkable responses, numerous patients still do not respond to these treatments (primary resistance), and some relapse after the initial response (acquired resistance)[4,5]. Alternate therapies are needed for patients with melanoma with primary or acquired resistance to immune checkpoint blockade therapy. Another relevant unmet medical need is the development of novel treatments for rare melanoma subtypes, including acral, mucosal, and uveal melanoma. These melanoma subtypes present at a lower incidence than cutaneous melanoma but tend to have poorer prognoses due to the late detection stage and the limited response to standard therapies. The overall response rate to CTLA-4 and PD-1 blockade combination therapy is

[1]Department of Hematology-Oncology, David Geffen School of Medicine at the University of California Los Angeles (UCLA), Los Angeles, CA, USA. [2]Parker Institute for Cancer Immunotherapy Center at UCLA, Los Angeles, CA, USA. [3]UCLA Stein Eye Institute and Department of Ophthalmology, David Geffen School of Medicine at UCLA, Los Angeles, CA, USA. [4]Division of Hematology/Oncology, VA Greater Los Angeles Healthcare System, Los Angeles, CA, USA. [5]Molecular Biology Institute, UCLA, Los Angeles, CA, USA. [6]Jonsson Comprehensive Cancer Center—UCLA, Los Angeles, CA, USA. [7]Division of Dermatology, Department of Medicine, UCLA, Los Angeles, CA, USA. [8]Department of Dermatology, VA Greater Los Angeles Healthcare System-West Los Angeles, Los Angeles, CA, USA. [9]Department of Microbiology, Immunology, and Molecular Genetics at UCLA, Los Angeles, CA, USA. [10]Broad Stem Cell Research Center—UCLA, Los Angeles, CA, USA. [11]These authors contributed equally: Sameeha Jilani, Justin D. Saco, Edurne Mugarza. ✉e-mail: CPuigSaus@mednet.ucla.edu

32% in acral[6], 23–37% in mucosal[6,7], and 15.6% in uveal melanoma[8]. In this work, we propose treating cutaneous and rare melanoma subtypes with chimeric antigen receptor (CAR)-T cell therapy. CAR-T cell therapy has shown remarkable antitumor responses in hematologic cancers, including lymphoma, leukemia, and multiple myeloma, leading to the FDA approval of CAR-T cell therapies targeting CD19 and BCMA[9,10]. However, treatment of solid tumors with CAR-T cell therapy has proven more challenging[11,12]. Several factors limit the success of CAR-T cell therapy in solid cancers, including CAR-T cell trafficking to the tumor, the immunosuppressive tumor microenvironment, and the lack of good target antigens expressed at high levels in the tumor but not in essential normal healthy tissues. Despite these challenges, early phase I clinical trials have shown success in treating patients with pediatric gliomas and neuroblastomas[13,14], and gastrointestinal cancers[15], among others, and the field is exploring further modifications to overcome the restrictive tumor microenvironment and improve tumor trafficking[16–18].

Tyrosinase-related protein 1 (TYRP1) is a transmembrane glycoprotein that plays an essential role in melanin synthesis. TYRP1 is one of the most abundant glycoproteins in both normal and malignant melanocytes, and its expression in normal tissues is limited to the skin and the retinal pigment epithelium (RPE)[19,20] (Human Protein Atlas). TYRP1 is primarily located intracellularly in the melanosomes, and at a lower frequency, on the plasma membrane[21–23]. Targeting cell surface expression of TYRP1 with monoclonal antibodies has demonstrated efficacy in preclinical models[24,25] and clinical trials[26]. A clinical trial using a monoclonal antibody (IMC-20D7S) targeting TYRP1 to treat patients with relapsed or refractory melanoma showed a disease control rate of 41%, with no adverse events reported[26]. This trial demonstrates the safety and feasibility of targeting TYRP1 surface expression in patients with melanoma and highlights the need to identify strategies to increase the overall antitumor activity. Monoclonal antibodies can directly induce apoptosis of tumor cells, antibody-dependent cell-mediated cytotoxicity, and activation of the complement cascade. CARs couple the antigen specificity of monoclonal antibodies with the effector functions of a T cell, leading to T-cell activation and cytotoxicity directly upon antigen detection. Moreover, CAR-T cells have improved biodistribution to the tumor compared to monoclonal antibodies and can expand and persist long-term in vivo, which avoids the need for repeated administrations[27]. Additionally, next-generation strategies for CAR-T cell engineering are being developed to i) improve recognition of heterogeneous antigens by targeting multiple antigens; ii) equip T cells with additional mechanisms to increase persistence in immunosuppressive environments; iii) enhance safety by adding kill switch strategies[12,28].

Using patient-derived models of cutaneous, acral, and uveal melanoma, we showed the relevance of TYRP1 surface expression as a target for CAR-T cell therapy for cutaneous melanoma and rare melanoma subtypes in patients with high TYRP1 overexpression. Applying a fine-tuning iterative design approach, we have built a CAR-T cell receptor that detects low levels of TYRP1 on the cell surface of the target cells, inducing CAR-T cell activation and robust antitumor activity in vitro and in vivo in all models tested. Our work represents an innovative approach to identifying CAR-T cell therapy targets based on aiming at intracellular transmembrane proteins that reach the cell surface through the secretory pathway. Our strategy could be extended to other protein targets in any cancer type. The success of our approach relies on the CAR design, which has been extensively optimized to recognize low levels of TYRP1 expression on the cell surface. Our results show that high TYRP1 overexpression is required to achieve detectable surface expression, allowing us to use the antigen density threshold as a mechanism to discriminate between tumor and normal tissues. The TYRP1 CAR-T cell therapy we have developed is exclusively designed to treat patients with melanoma. Based on the efficacy, selectivity, and toxicity results presented in this manuscript, we are currently planning the clinical translation of this therapy.

## Results

### Surface expression of TYRP1 is a relevant target for CAR-T cell therapy for patients with melanoma

Melanoma differentiation antigens (MDAs) are attractive targets for the design of melanoma treatments. MDAs include, among others, glycoprotein 100 (gp100), melanoma antigen recognized by T cells-1 (MART-1), tyrosinase, and TYRP1, which are involved in the melanin biosynthesis pathway. The melanin synthesis pathway is activated in all melanocytes in the skin, the RPE, and the inner ear and is overexpressed in a subset of patients with melanoma[29]. We combined the transcriptional profile of 723 lesions in a cohort that includes all the patients with melanoma in The Cancer Genome Atlas (TCGA) program, the CheckMate 038[30], and Keynote-001 (MK3475-001)[31] clinical trials. In this cohort, around 70% of all melanomas express *TYRP1* (Log2 FPKM (Fragments Per Kilobase of transcript per Million mapped reads) ≥1, Fig. 1a), and ~30% of them present high overexpression (Log2 FPKM ≥ 7, Fig. 1a). ICB is the standard of care for patients with cutaneous melanoma. Therefore, a potential therapy targeting TYRP1 would be administered to patients who do not respond or relapse after initial treatment with ICB. To further explore the clinical relevance of this target, we quantified the *TYRP1* expression before and after ICB therapy in patients with response (R), no response (NR), or stable disease (SD). Treatment with ICB does not significantly change *TYRP1* expression in any of the three groups (Fig. 1b). Among the 37 patients without response to ICB, 65% of biopsies after treatment express *TYRP1* (Log2 FPKM ≥1), and 35% present high overexpression (Log2 FPKM ≥ 7) (Fig. 1c) and would potentially benefit from a therapy targeting TYRP1. Of note, 52% of the 29 patients with response to therapy express *TYRP1* (Log2 FPKM ≥1) in the on-treatment biopsies, and 21% have overexpression (Log2 FPKM ≥ 7) (Fig. 1c). Interestingly, 59%, 60%, and 92% of patients with acral, mucosal, and uveal melanoma, respectively, present high expression of *TYRP1* (Log2 FPKM ≥ 7, Fig. 1d–f) and could benefit significantly from a T-cell therapy designed to target TYRP1. Finally, we measured the TYRP1 protein level by immunohistochemistry in metastatic lesions of cutaneous (*n* = 11), acral (*n* = 10), mucosal (*n* = 6), and uveal (*n* = 11) melanoma and showed that a significant subset of the patients analyzed presented high and homogeneous TYRP1 expression (Fig. 1g, h and Supplementary Fig. 1a). We validated the staining with a TYRP1-specific antibody and used the 38 patient samples from metastatic melanoma lesions to develop a scoring system that will be used as an inclusion-exclusion criterion to enroll patients in the phase I clinical trial. Since the TYRP1 CAR-T cells target only melanoma cells with high levels of TYRP1 expression, our scoring method will only consider the percentage of cells with a 3+ intensity stain. Using this system and a cut-off of 65% of total tumor cells with 3 + TYRP1 stain, 11 of the 38 patients screened would be eligible for the clinical trial (~29% of patients, Fig. 1h, dashed line). By reducing the criteria to 50% of total tumor cells with 3 + TYRP1 stain, 15 of the 38 patients would be eligible, ~40% of all screened patients (Fig. 1h, black line).

To further investigate the relevance of TYRP1 as a T-cell therapy target for melanoma treatment, we measured the expression levels by RNAseq in a panel of 54 patient-derived melanoma cell lines[32]. Similar to the results observed in melanoma lesions, 27 of these cell lines expressed *TYRP1* (Log2 FPKM ≥1), and 15 of them presented a high overexpression (Log2 FPKM ≥ 7, Fig. 1i). It is relevant to highlight that we set up the cut-off of positive and high *TYRP1* expression based on experimental data generated using cell lines and applied the same cut-off for cell lines and tumors. However, RNAseq from bulk tumors underrepresents the tumor content compared to cell lines and may underestimate the number of patients with TYRP1 overexpression. We

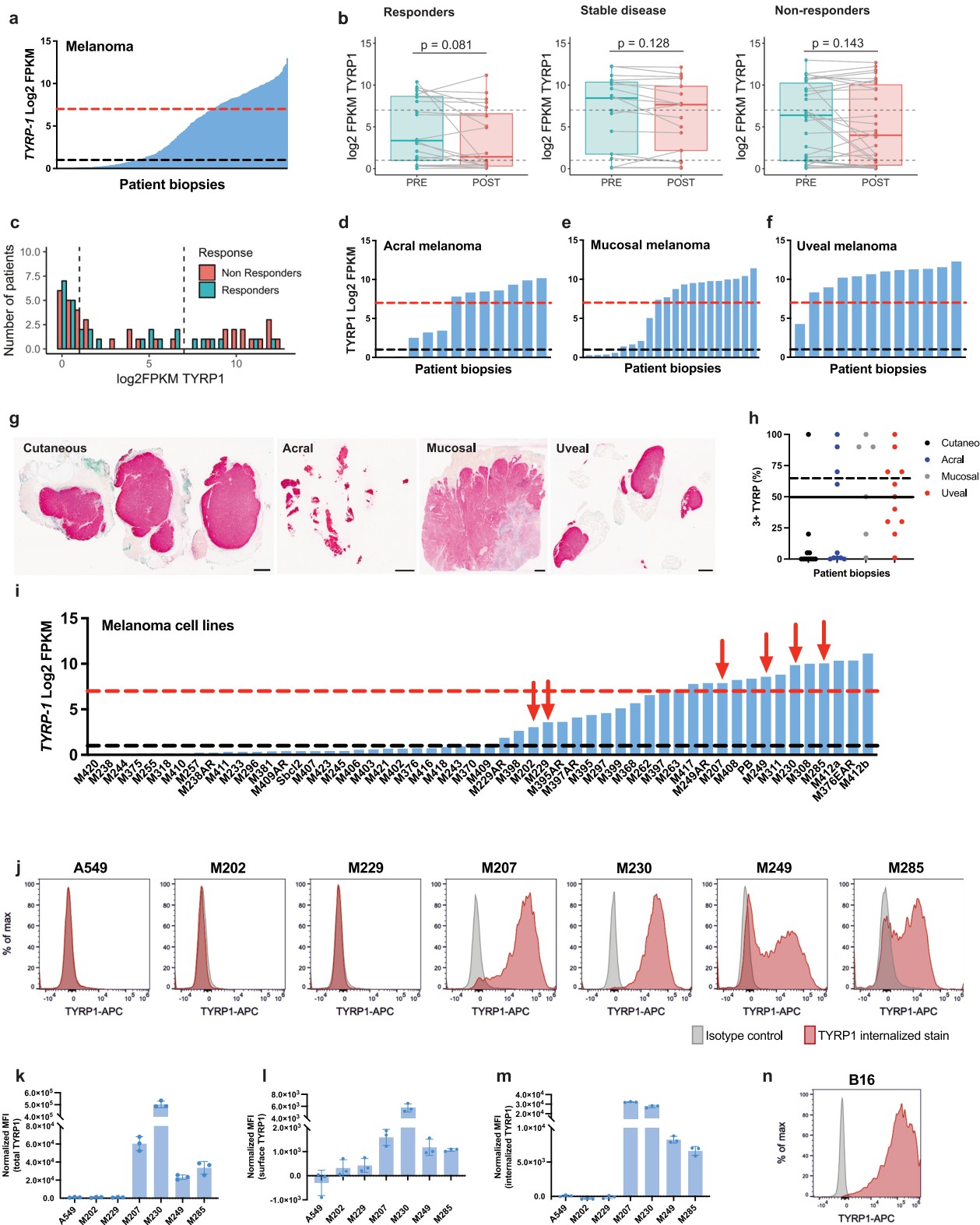

next selected a reduced panel of patient-derived cell lines with high (M285, M249, M230, and M207), intermediate (M229 and M202), and negative (A549, a lung adenocarcinoma cell line) expression of *TYRP1* measured by RNAseq and quantified the intracellular and surface expression TYRP1 by flow cytometry. As a result of the fusion of the melanosomes with the plasma membrane, TYRP1 is transported to the cell surface, and it remains on the surface until it is recycled and endocytosed. In order to account for the transitory membrane

expression of TYRP1, we quantified the total amount of TYRP1 on the surface and internalized during a window of 16h. To this end, we incubated cells in culture with the TYRP1 antibody and monensin, which blocks the degradation of internalized proteins, for 16h. In this manner, we sought to quantify surface TYRP1 over the entire incubation period, even if it was internalized during this time. We were able to detect surface expression of TYRP1 in cell lines with high expression of *TYRP1*, including patient-derived cell lines and the murine B16

**Fig. 1 | Surface expression of TYRP1 is a relevant target for CAR-T cell therapy for patients with melanoma. a** *TYRP1* expression in melanoma biopsies combining the TCGA melanoma ($n = 472$), the CheckMate038 ($n = 191$), and the Keynote-001 ($n = 60$) clinical trial datasets ($n = 723$). Expression measured by RNAseq. Black line positive *TYRP1* expression (≥1 Log2 FPKM). Red line high *TYRP1* expression (≥7 Log2 FPKM). **b** *TYRP1* Log2 FPKMs of paired baseline and on-treatment biopsies from the CheckMate 038 clinical trial were classified based on clinical response as responders ($n = 20$), stable disease ($n = 17$), and non-responders ($n = 30$). Responders defined as CR and PR and non-responders are defined as PD. Two-sided, paired *t* test *p* values are shown. Data is presented in Boxplot format where the center is the median and top and bottom bounds represent 75th and 25th percentile, respectively. **c** Histogram showing the number of patients with negative (≤1 Log2 FPKM), intermediate, and high (≥7 Log2 FPKM) *TYRP1* expression in on-treatment samples after receiving ICB (CheckMate 038 clinical trial) classified based on response. SD are excluded from this analysis, R and NR patients as in (**b**). **d**–**f** *TYRP1* expression in acral (**d**, $n = 12$), mucosal (**e**, $n = 20$), and uveal melanoma (**f**, $n = 12$). Same dataset and cutoff as (**a**). **g** TYRP1 expression measured by immunohistochemistry stain in metastatic lesions of cutaneous, acral, mucosal, and uveal melanoma. Scale bar equal to 2 mm. One representative image of cutaneous ($n = 11$), acral ($n = 10$), mucosal ($n = 6$), and uveal ($n = 11$) melanoma is shown. **h** TYRP1 score in metastatic lesion from patients with cutaneous ($n = 11$), acral ($n = 10$), mucosal ($n = 6$), and uveal ($n = 11$) melanoma. The solid and dashed black lines indicate 50% and 65% of the total tumor cells with 3 + TYRP1 stain, respectively. **i** *TYRP1* expression in a panel of 54 patient-derived melanoma cell lines. Expression measured by RNAseq. Same cutoff as (**a**). Red arrows indicate cell lines used in subsequent experiments. **j** Expression of surface and internalized *TYRP1* after 16 h culture with the TYRP1 antibody (red) compared to the isotype control (gray) in cell lines with high (≥7 Log2 FPKM), intermediate (≥1 Log2 FPKM) and negative (<1 Log2 FPKM) *TYRP1* RNA levels. One representative sample of three replicates is shown. Total (**k**), surface only (**l**), and surface/internalized over a 16 h period (**m**) TYRP1 expression normalized by the isotype control in the same cell lines (mean ± SD, $n = 3$ independent samples). **n** Expression of surface/internalized *TYRP1* after 16 h culture with the TYRP1 antibody (red) compared to the isotype control (gray) in B16 cells. One representative sample of three replicates is shown. FPKM Fragments Per Kilobase of transcript per Million mapped reads. CR complete response, PR partial response, PD progressive disease, SD stable disease. Source data and exact *p* values are provided as a Source Data file.

melanoma model, and we showed that the total intracellular, internalized, and surface expression correlated with the level of expression measured by RNAseq (Fig. 1 j–n and Supplementary Fig. 1b–d). Similar results were obtained using the 20D7S antibody to detect surface, internalized, and total TYRP1 (Supplementary Fig. 1c, d). Quantification of the surface and internalized TYRP1 confirms the cycling nature of TYRP1. Finally, we performed immunohistochemistry (IHC) staining using the same antibody optimized for the patient biopsies. Expression patterns were comparable across flow cytometry and IHC, with some cell lines (such as M230) exhibiting homogeneous expression, while others like M249 showed a bimodal distribution (Supplementary Fig. 1e). Altogether, these results highlight the clinical relevance of TYRP1 as a target for melanoma therapies and demonstrate the feasibility of targeting the surface expression of TYRP1 with CAR-T cell therapy.

## TYRP1 CAR-T cell therapy optimized to detect low levels of TYRP1 on the cell surface leads to cytotoxicity and cytokine release against patient-derived and murine melanoma models

The 20D7S is a fully human antibody that recognizes TYRP1 with an affinity of 1.5 nM[33]. This antibody has been previously used as a single agent in clinical trials for relapsed or refractory melanoma showing moderate antitumor activity and a lack of severe adverse events related to the treatment[26]. Based on these results, we selected the 20D7S antibody clone to design the TYRP1 CAR. Using this single chain variable fragment (scFv), we constructed second-generation CARs with the CD28 transmembrane domain and the signaling domains of the CD28 costimulatory molecule, and the CD3zeta T-cell co-receptor. The hinge connects the antigen binding domain to the transmembrane domain, and its length and sequence determine the flexibility of the CAR to access the antigen. Long spacers facilitate binding to epitopes that are proximal to the plasma membrane of the target cell, while shorter hinges are more effective at binding membrane-distal epitopes[34]. We optimized the hinge length to maximize the productive interaction with the antigen and the antitumor activity. We tested three CAR constructs with IgG-based hinges of different sizes containing the IgG4 hinge alone (12 amino acids) or together with the IgG4-CH3 (107 amino acids) or the IgG4-CH2-CH3 (217 amino acids) domains. These three constructs were named 20D7SS-28ζ, 20D7SM-28ζ, and 20D7SL-28ζ, respectively. Of note, we introduced the L235E and the N297Q mutations in the CH2 region to reduce binding to Fc gamma receptors (FcγRs) and increase CAR-T cell persistence[35,36] (Fig. 2a). We cloned the three constructs in an MSGV-1 retroviral vector that expresses the CAR under the control of the MSCV (murine stem cell virus) promoter and has been splicing optimized[37]. The three CAR constructs fold correctly

and are transported to the cell surface when expressed in human primary T cells (Fig. 2b). Next, we measured the cytotoxicity and cytokine release of the 20D7S-derived CARs when co-cultured with a panel of patient-derived melanoma cell lines with high (M285, M249, M230, and M207) and intermediate (M202 and M229) total TYRP1 expression. Our results show that all constructs lead to target cell killing and interferon-gamma (IFNγ) release upon co-culture with melanoma cells with high expression of total TYRP1 but not when co-cultured with cells with intermediate expression (Fig. 2c, d). In this setting, the IFNγ release increases with the length of the hinge. In dose-response cytotoxicity studies, at 48h after co-culture, the 20D7SM and 20D7SL outperformed 20D7SS, showing cytotoxicity in TYRP1[high] melanoma cell lines even at the lowest dose (Fig. 2e). Of note, 20D7SM-28ζ CAR surface expression is slightly lower than the other two constructs (Fig. 2b), which may also play a role in the overall CAR-T cell activity. Additionally, moderate-to-low cytotoxicity was observed with the three constructs in TYRP1[intermediate] cell lines only at the highest effector-to-target ratio (Fig. 2e). These results suggest that only tumor cell lines with high expression of total TYRP1 present sufficient surface protein to activate the CAR-T cells. This antigen density requirement will enhance the selectivity of the TYRP1 CAR against highly pigmented tumors with high TYRP1 expression, sparing the normal cell in the skin and the retina. As a result, the M202 cell line that did not present enough TYRP1 on the cell surface to activate TYRP1 CAR-T cell killing, was efficiently killed by T cells expressing a MART1 TCR, an antigen co-expressed with TYRP1 (Supplementary Fig. 2a).

Importantly, the epitope recognized by the 20D7S antibody is conserved between the human and murine TYRP1 protein, allowing evaluation of the human CAR construct against syngeneic tumor models. Indeed, the three 20D7S-derived TYRP1 CARs are cytotoxic and secrete cytokines upon co-culture with the B16 murine melanoma cell line (Fig. 2f–h, Supplementary Fig. 2b).

## TYRP1 CAR with a long flexible hinge exhibits superior tumor control in TYRP1[high] syngeneic and patient-derived melanoma models

Upon establishing the function of 20D7S-based TYRP1 CAR-T cells in vitro, we assessed their antitumor activity in vivo. First, we measured the ability of the 20D7SS-28ζ, 20D7SM-28ζ, and 20D7SL-28ζ murine CAR-T cells to control tumor growth in immunocompetent C57BL/6 mice bearing B16 murine melanoma tumors. We adoptively transferred $5 × 10^6$ murine 20D7S-derived CAR-T cells with three doses of interleukin-2 (IL-2) on consecutive days to boost CAR-T cell proliferation. Prior to T-cell administration, mice received 5Gy total body irradiation to allow the engraftment of the transferred T cells (Fig. 3a). We

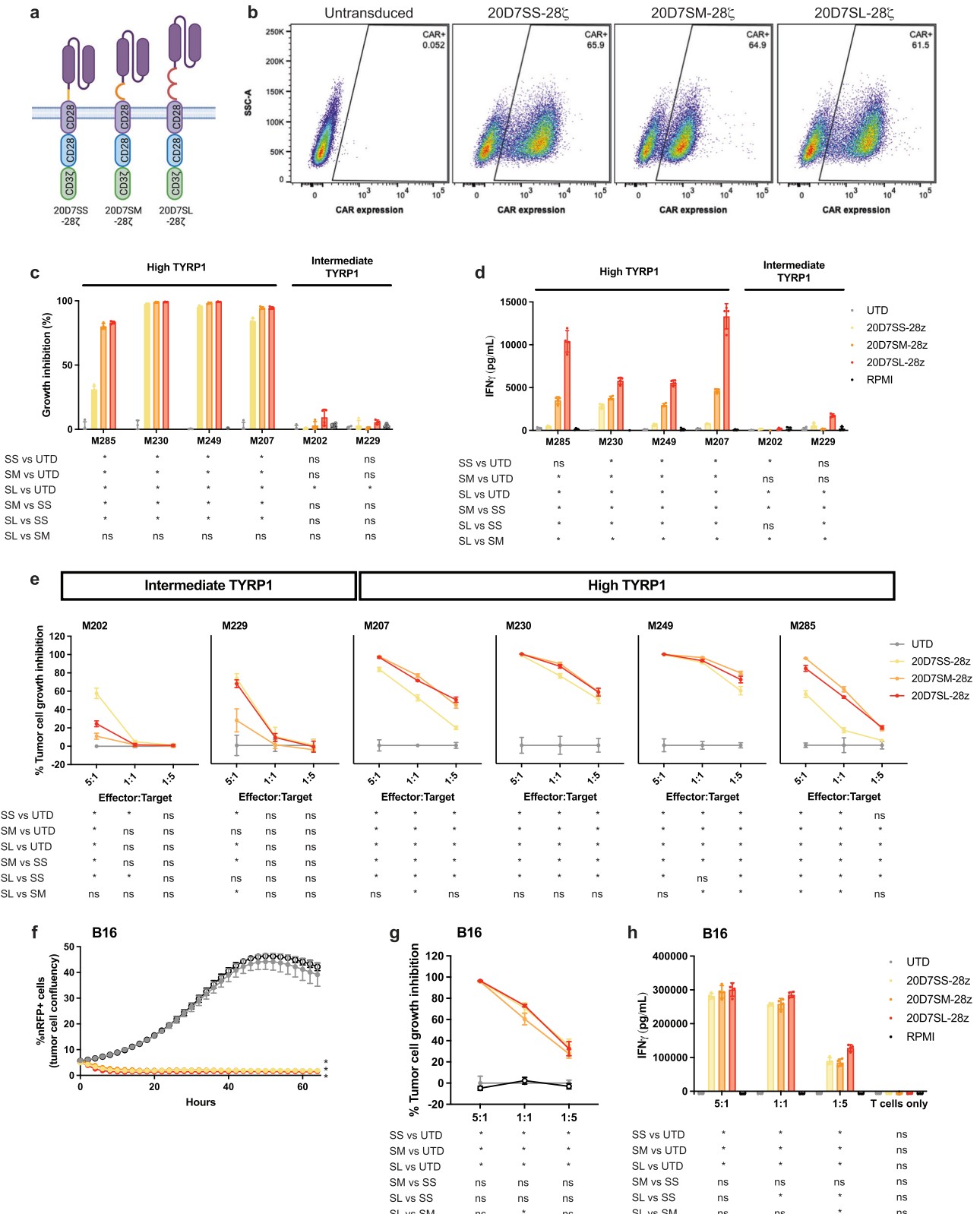

followed tumor growth and survival over time and observed that, despite the relatively similar activity shown by these three CAR constructs in vitro, only T cells expressing the 20D7SL-28ζ CAR—the CAR with the longest hinge—were able to control tumor growth and improve mouse survival (Fig. 3b, c; Supplementary Fig. 2c, d). Based on these results, we selected the 20D7SL-28ζ CAR-T cells for further investigation. Of note, the level of CAR expression is lower in the

20D7SM-28ζ CAR construct, which may also play a role in the overall CAR-T cell activity (Supplementary Fig. 2b).

We next assessed the effect of the 20D7SL-28ζ CAR-T cell dose on the overall antitumor activity and observed no significant differences in tumor growth control and survival comparing the administration of $3.5 \times 10^6$ and $7 \times 10^6$ CAR$^+$ T cells (Fig. 3d–f, Supplementary Fig. 2e, f). Adoptive T-cell therapy protocols for solid tumors frequently include

**Fig. 2 | TYRP1 CAR-T cell therapy optimized to detect the low levels of TYRP1 on the cell surface leads to cytotoxicity and cytokine release against patient-derived and murine melanoma models. a** Schematics of the 20D7SS-28ζ, 20D7SM-28ζ, and the 20D7SL-28ζ CAR-T cells. Graphical depictions were created with BioRender.com. **b** Human primary T cells transduced with the TYRP1 CARs. Representative flow cytometry plots showing CAR expression on the cell surface. CAR expression was detected with the Kip-1 anti-whitlow linker antibody. Representative flow cytometry showing transduction efficiency of one experiment out of two experiments performed using these constructs. **c, d** Co-culture of 20D7SS-28ζ, 20D7SM-28ζ, and the 20D7SL-28ζ with a panel of TYRP1[high] and TYRP1[intermediate] patient-derived cell lines. Mean ± SD (n = 4, independent co-cultures) are plotted. **c** Percentage of tumor cell growth inhibition normalized by the growth of the cell lines co-cultured with untransduced T cells at 96 h after co-culture at a 1:1 effector to target (E:T) ratio. **d** IFNγ released at 24 h after co-culture measured by ELISA (5:1, E:T ratio). **e** Cytotoxicity dose–response curves of 20D7SS-28ζ, 20D7SM-28ζ, and the 20D7SL-28ζ CAR-T cells upon 48 h co-culture with a panel of TYRP1[high] and TYRP1[intermediate] melanoma cell lines. Percentage of tumor cell growth inhibition normalized by the untransduced T cells control. Mean ± SD (n = 4, independent co-cultures) are plotted. **f–h** Antitumor activity of the 20D7SS-28ζ, 20D7SM-28ζ, and the 20D7SL-28ζ murine CAR-T cells upon co-culture with the B16 murine melanoma cell line. Untransduced murine T cells, or media alone are used as negative controls. Mean ± SD (n = 4, independent co-cultures) are plotted. **f** Time-course of B16 murine melanoma tumor cell growth and inhibition (5:1 E:T). **g** Cytotoxicity dose-response curves at 48 h after co-culture. **h** Dose–response IFNγ release at 24 h after co-culture. Cytokine secretion of T cells alone without stimulation is shown as a negative control. *p < 0.05 unpaired, two-tailed t test with Holm-Sidak adjustment for multiple comparisons. Source data and exact p values are provided as a Source Data file.

IL-2 to boost the expansion of the adoptively transferred T cells. However, a large fraction of the toxicities seen in early clinical trials were associated with the high-dose IL-2 treatment[38,39]. Interestingly, we showed that murine 20D7SL CAR-T cells effectively control tumor growth with or without IL-2 supplement (Fig. 3g–i, Supplementary Fig. 2g–h), even in the highly aggressive B16 murine tumor model that does not respond to PD-1 blockade due to immune exclusion mechanisms[31]. These results demonstrate that adoptive T-cell therapy can be successful even in non-immunogenic tumors with a low-to-negative response to ICB, provided that the targeted antigen is expressed at sufficient levels and the CAR construct is optimized to maximize antigen detection.

Finally, we studied the antitumor activity of the 20D7SL-28ζ CAR-T cells in patient-derived cutaneous melanoma models in immuno-deficient mice. Briefly, M207 (Fig. 3j) and M249 (Fig. 3l) cells were implanted subcutaneously in the flanks of NOD/Scid IL-2RγC-null (NSG) mice. Once the tumors engrafted, 5–6x10[6] human 20D7SL-28ζ CAR-T cells were administered systemically, and the tumor growth was followed until mice showed signs of graft-versus-host disease (GVHD) and all groups had to be euthanized. CD19 CAR-T cells were used as a negative control. In these models, the 20D7SL-28ζ CAR-T cells were able to control tumor growth and induce tumor regression (90–100% complete responses (CR), (Fig. 3j–m; Supplementary Fig. 2i–l). Importantly, no major signs of toxicity associated with this treatment were observed.

To confirm CAR-T cell expansion in 20D7SL-treated tumors, we quantified T-cell infiltration in the tumor. While no T cells were detected in UTD-treated B16 tumors, 20D7SL-28ζ–treated tumors had an increased T-cell infiltration (Fig. 3n, Supplementary Fig. 10a). Similarly, in the human model M207, we were able to detect human CD45 + T cells only in tumors treated with 20D7SL-28ζ (Fig. 3o, Supplementary Fig. 10b). Interestingly, we found hCD45[+] cells in spleens of UTD-treated mice but not in tumors (Fig. 3p, Supplementary Fig. 10b), suggesting that although adoptively transferred T cells survive in the periphery, only CAR[+] T cells are able to traffic into tumors.

## CD28 outperforms 4-1BB as the costimulatory signal for the 20D7SL CAR, leading to antitumor activity in a wider tumor panel

To facilitate clinical translation, we further evaluated the 20D7SL CAR in a lentiviral vector expression system due to its enhanced safety profile. Our therapeutic candidate uses an epHIV7-based lentiviral vector that places the expression of the 20D7SL CAR under the control of the human EF1α (Translation Elongation Factor 1-α) promoter. This vector is currently being used in several clinical trials (NCT02208362, NCT04003649, NCT04119024, NCT02051257, and NCT04007029, among others). To further optimize the TYRP1 CAR clinical candidate, we studied the effect of costimulatory signaling in TYRP1 CAR-T cells' antitumor activity. We compared T cells that express 20D7SL CAR constructs differing only in the costimulatory signaling domain,

comprising either CD28 (20D7SL-28ζ) or 4-1BB (20D7SL-BBζ) (Fig. 4a). In vitro, in TYRP1[high] patient-derived melanoma models, 20D7SL-28ζ and 20D7SL-BBζ CAR-T cells present equivalent cytotoxicity (Supplementary Fig. 3a) and cytokine secretion (Fig. 4b). However, under stress-test conditions in which the CAR-T cells were exposed to ten cycles of antigen challenge using two patient-derived TYRP1[high] melanoma cell lines, 20D7SL-28ζ CAR-T cells consistently outperformed 20D7SL-BBζ CAR-T cells (Fig. 4c,d). In in vivo models of human melanoma xenografts in immunodeficient NSG mice, 20D7SL-28ζ CAR-T cells showed antitumor activity against M249 and M230 tumors, with more efficient control observed in the M249 model. In contrast, 20D7SL-BBζ CAR-T cells had a relatively weak effect on M249 tumors but cleared M230 tumors (Fig. 4e, f; Supplementary Fig. 3b, c). We assessed TYRP1 expression in tumors after treatment with the TYRP1 CAR (Supplementary Fig. 3d). M230 tumors, consistent with our in vitro data (Fig. 1j) present a homogeneous expression pattern of TYRP1. They present lower response to 20D7SL-28z and TYRP1 expression is maintained after treatment. In contrast, 20D7SL-BBz completely eliminated M230 tumors in 4/5 mice. We analyzed the residual tissue in the remaining mouse and found patchy or low TYRP1 expression, which could be due to an antigen loss (Supplementary Fig. 3d, top). Conversely, M249 tumors, which show heterogeneous expression of TYRP1, respond well to 20D7SL-28z treatment and 3/8 tumors are eliminated at the time of the experiment endpoint. TYRP1 expression in residual tumors is absent, suggesting growth of TYRP1[neg] clones after treatment. However, with 20D7SL-BBz CAR, tumor control is more modest and TYRP1 expression is maintained in the tumors, highlighting the presence of other resistance mechanisms (Supplementary Fig. 3d, bottom). The differences in antitumor activity between the 20D7SL-28ζ and the 20D7SL-BBζ are likely associated with the antigen density. CAR constructs with the 4-1BB costimulatory domain are more sensitive to the antigen threshold with decreased activities in lower antigen density tumors[40]. Of note, these two studies were conducted in parallel, using the same CAR-T cell products.

## 20D7SL-28ζ and 20D7SL-BBζ CAR-T cells specifically recognize TYRP1

To confirm the antigen specificity of the 20D7SL-derived TYRP1 CAR-T cell therapy, we next knocked out the TYRP1 protein in three patient-derived TYRP1[high] melanoma cell lines (M207, M249, and M285, Supplementary Fig. 4a–e) and measured cytotoxicity and cytokine release upon co-culture of 20D7SL-28ζ and 20D7SL-BBζ CAR-T cell with the parental TYRP1[high] cell lines or their *TYRP1*-knock out counterparts. Untransduced T cells were used as controls. We showed that 96 hours after co-culture, TYRP1 CAR-T cells completely inhibited the growth of the three parental tumor cell lines, but this activity was lost in the *TYRP1*-knock-out cell lines (Fig. 5a). Moreover, both 20D7SL-28ζ and the 20D7SL-BBζ CAR-T cells secrete IFNγ upon co-culture with the parental cell lines, but this activity was absent upon co-culture with the *TYRP1*-knock out cell lines (Fig. 5b). Cytotoxicity dose response at

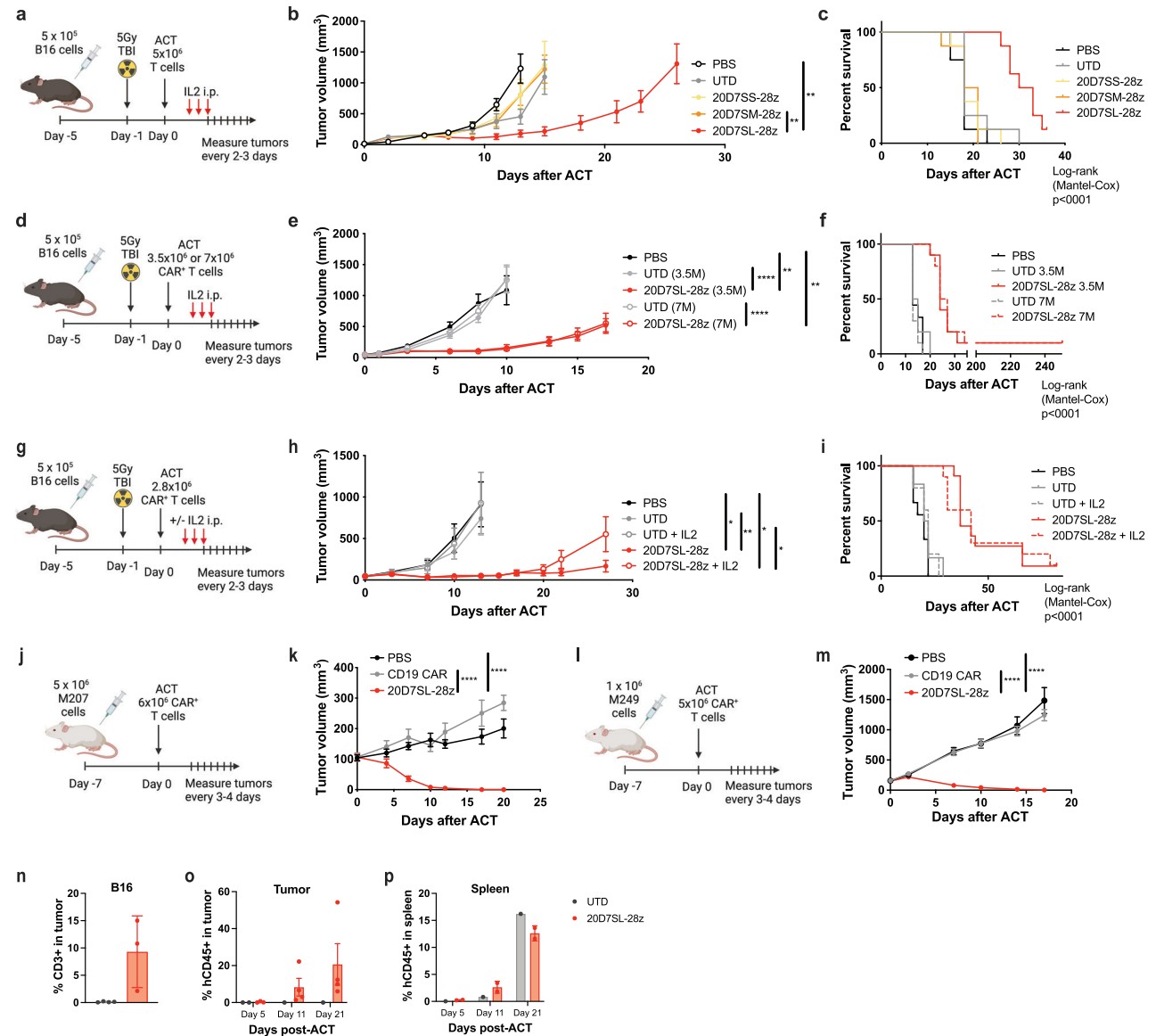

**Fig. 3 | TYRP1 CAR with a long flexible hinge exhibits superior tumor control in TYRP1^high syngeneic and patient-derived melanoma models.**
**a, d, g, j, l** Schematics of the in vivo mouse studies indicating the timeline, tumor cell and CAR-T cell doses, and irradiation and IL-2 doses and timelines if applicable. Graphical depictions were created with BioRender.com. **b, e, h, k, m** Kinetic of tumor growth or regression over time after treatment with 20D7S-derived CAR-T cells alone or in combination with IL-2. Untransduced T cells or CD19 CAR-T cells and vehicle were used as controls. Mean ± SEM are plotted (**b**: $n = 8$, **e**: $n = 10$, **h**: $n = 10$ for 20D7SL and 20D7SL + IL2, $n = 6$ for UTD, $n = 5$ for UTD + IL2, $n = 9$ for PBS, **k**: $n = 10$, **m**: $n = 10$). * $p < 0.05$, ** $p < 0.005$, **** $p < 0.0005$ unpaired, two-tailed $t$ test

with Holm-Sidak adjustment for multiple comparisons. **c, f, i** Kaplan-Meier survival curves. Survival differences are statistically significant. Log-rank (Mantel-Cox) $p < 0.0001$. **n** Percentage of CD3+ cells from single cells in B16 tumors treated with 20D7SL-28z or untransduced CAR-T cells on Day 13 after ACT. Mean ± SEM is plotted ($n = 4$ for UTD, $n = 3$ for 20D7SL-28z). Percentage of CD45^+ from single cells in tumors (**o**) and spleens (**p**) of M207-bearing mice treated with 20D7SL-28z or untransduced CAR-T cells at day 5, 11, and 21 after ACT. Mean ± SEM are plotted (in (**o**), $n = 2$ for UTD day 5, $n = 1$ for UTD days 11 and 21, $n = 3$ for 20D7SL-28z day 5 and $n = 4$ for 20D7SL-28z days 11 and 21. In (**p**), $n = 1$ for UTD and $n = 2$ for 20D7SL-28z at all time points). Source data and exact $p$ values are provided as a Source Data file.

48 hours after co-culture confirmed the loss of cytotoxicity of the TYRP1 CAR against the *TYRP1*-knockout cell lines (Fig. 5c). To further confirm TYRP1 CAR selectivity, we assessed the cytotoxicity and cytokine release upon co-culture with a panel of non-melanoma TYRP1^neg cell lines (A549 lung adenocarcinoma as well as UPS03 and UPS04 undifferentiated pleomorphic sarcoma). Both the 20D7SL-28ζ and the 20D7SL-BBζ CAR-T cells show no reactivity against these TYRP1^neg cell lines, confirming the antigen specificity of the CAR-T cells (Fig. 5d, e).

To further explore the specificity of the 20D7S-derived CAR-T cell therapy, we characterized the binding of the 20D7S antibody to a panel of 35 human tissues derived from three different donors. These studies

showed extracellular staining within the kidney tubules, which was considered low toxicologic risk by the pathologist that conducted this study. In this study, extracellular TYRP1 staining was not observed in any other tissue. The overall goal of this study was to identify tissues that could suffer on-target off-tumor toxicities. The potential off-tumor activity identified in the kidney was deemed low risk, but we will monitor kidney function as part of the safety endpoints in the clinical trial.

**20D7SL-28ζ CAR-T cells exert strong tumor control and lack toxicity in the immunocompetent B16 melanoma model**
The most prevalent toxicities induced by CAR-T cell therapy are associated with the lymphodepleting pre-conditioning and cytokine

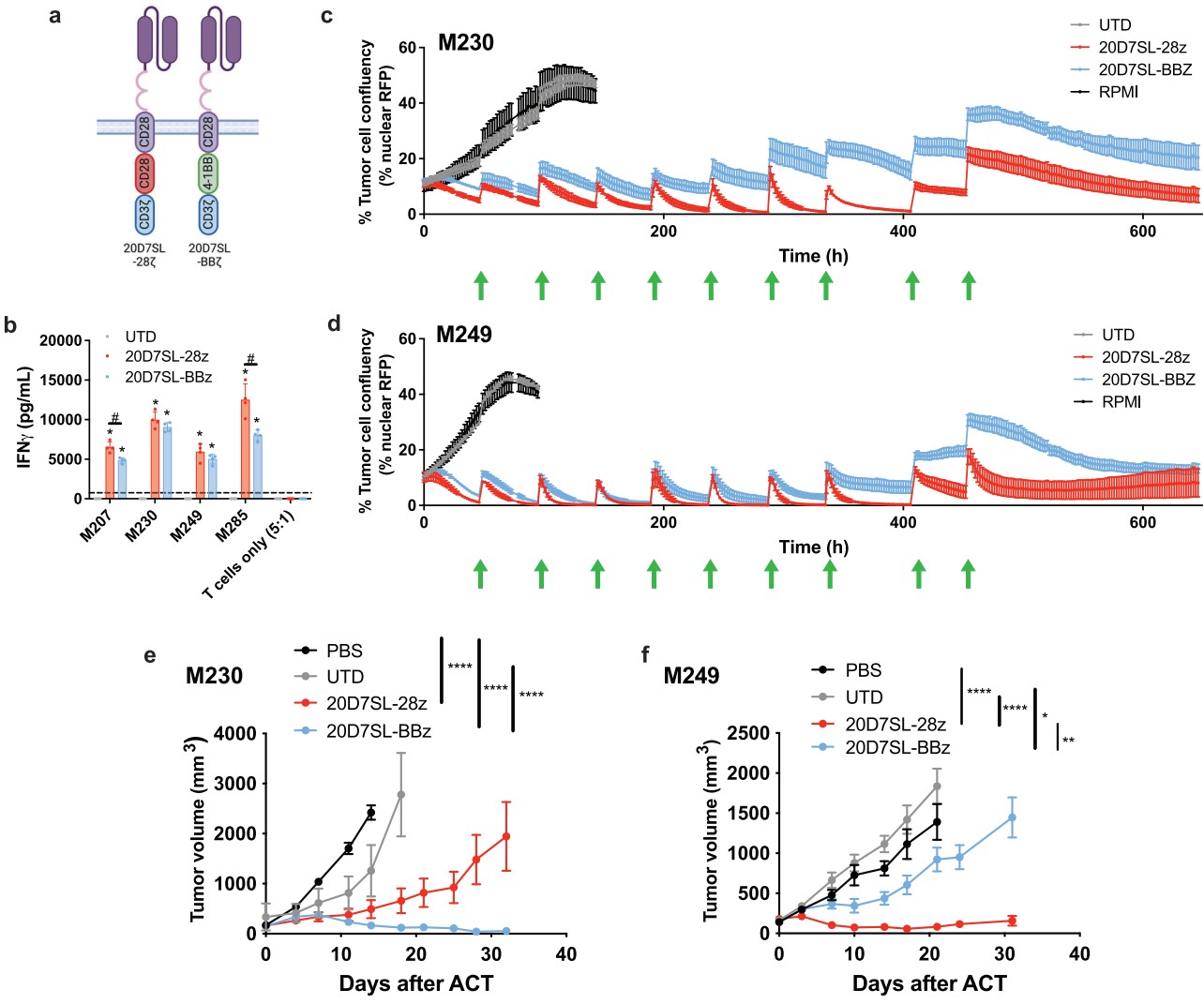

**Fig. 4 | CD28 outperforms 4-1BB as the costimulatory signal for the 20D7SL CAR, leading to antitumor activity in a larger tumor panel. a** Schematics of the 20D7SL-28ζ and the 20D7SL-BBζ CAR-T cells. Graphical depictions were created with BioRender.com. **b** IFNγ measured by ELISA 24 h after co-culture 20D7SL-28ζ and 20D7SL-BBζ CAR-T cells with a panel of TYRP^high patient-derived melanoma cells (5:1, E:T ratio). Untransduced T cells (UTD) were used as a negative control. Mean ± SD are plotted (*n* = 4, independent co-cultures). *p < 0.05 20D7SL-28z vs UTD, #p < 0.05 20D7SL-28z vs 20D7SL-BBζ, unpaired, two-tailed *t* test with Holm-Sidak adjustment for multiple comparisons. Percentage of tumor cell confluency over time upon co-culture of 20D7SL-28ζ and 20D7SL-BBζ CAR-T cells with ten consecutive antigen challenges with M230 (**c**) or M249 (**d**) melanoma cells. Untransduced T cells (UTD) and media alone were used as a negative control during

the first two challenges. Mean ± SD are plotted (*n* = 4, independent co-cultures). The green arrow in the *x*-axis marks each challenge. Kinetic of tumor growth or regression over time after treatment with 20D7SL-28ζ and the 20D7SL-BBζ CAR-T cells in NSG mice bearing M230 (**e**) and M249 (**f**) subcutaneous tumors. Untransduced T cells and PBS were used as controls. Tumor size was measured with a caliper. Mean ± SEM are plotted (**e:** *n* = 10 mice for PBS and 20D7SL-BBz, *n* = 2 for UTD, *n* = 8 for 20D7SL-28z; **f** *n* = 8 for PBS and 20D7SL-28z, *n* = 10 for UTD and 20D7SL-BBz). Representative of two independent experiments. *p < 0.05, **p < 0.005, ****p < 0.0005 unpaired, two-tailed *t* test with Holm-Sidak adjustment for multiple comparisons. Source data and exact *p* values are provided as a Source Data file.

release syndrome, a systemic inflammation caused by the cytokines released by the CAR-T cells. Such systemic toxicities are difficult to assess in immunodeficient NSG mice. 20D7SL CAR's cross-reactivity against human and murine TYRP1 allows us to evaluate toxicity in the syngeneic B16 melanoma model in immunocompetent C57BL/6 mice subjected to lymphodepleting total body irradiation one day prior to CAR-T cell administration. Consistent with previous observations (Fig. 3a–i), treatment with 20D7SL-28ζ CAR-T cells controlled B16 tumor growth (Fig. 6a, Supplementary Fig. 5a). There were no statistically significant changes in the body weight and temperature among untreated mice or those treated with either 20D7SL-28ζ CAR-T cells or untransduced T cells (Supplementary Figs. 5b, c). Of note, untreated mice were not irradiated and did not receive T cells. CAR-T cell persistence was measured in the tumor and spleen of mice from all groups

on days 5 and 13 after T-cell transfer and at the experiment endpoint on day 27 only for the CAR-T cell-treated group. Untreated mice and mice receiving untransduced T cells were euthanized on day 13 due to the large tumor size. CAR-T cell persistence was assessed by vector copy number quantification using digital droplet PCR with primers specific for the viral vector (Fig. 6b, c). Complete blood cell counts and blood chemistry were performed at the same time points, with results indicating a decrease in the white cell blood counts, hemoglobin, hematocrit, and platelets on day five after T-cell transfer in mice receiving total body irradiation together with untransduced T cells or 20D7SL-28ζ CAR-T cells. Red blood cell counts were recovered in both groups by day 13 and platelet counts by day 27, but white blood cell counts did not reach the level of untreated mice by the experiment endpoint (Fig. 6d–g). No differences were observed in the blood cell differential

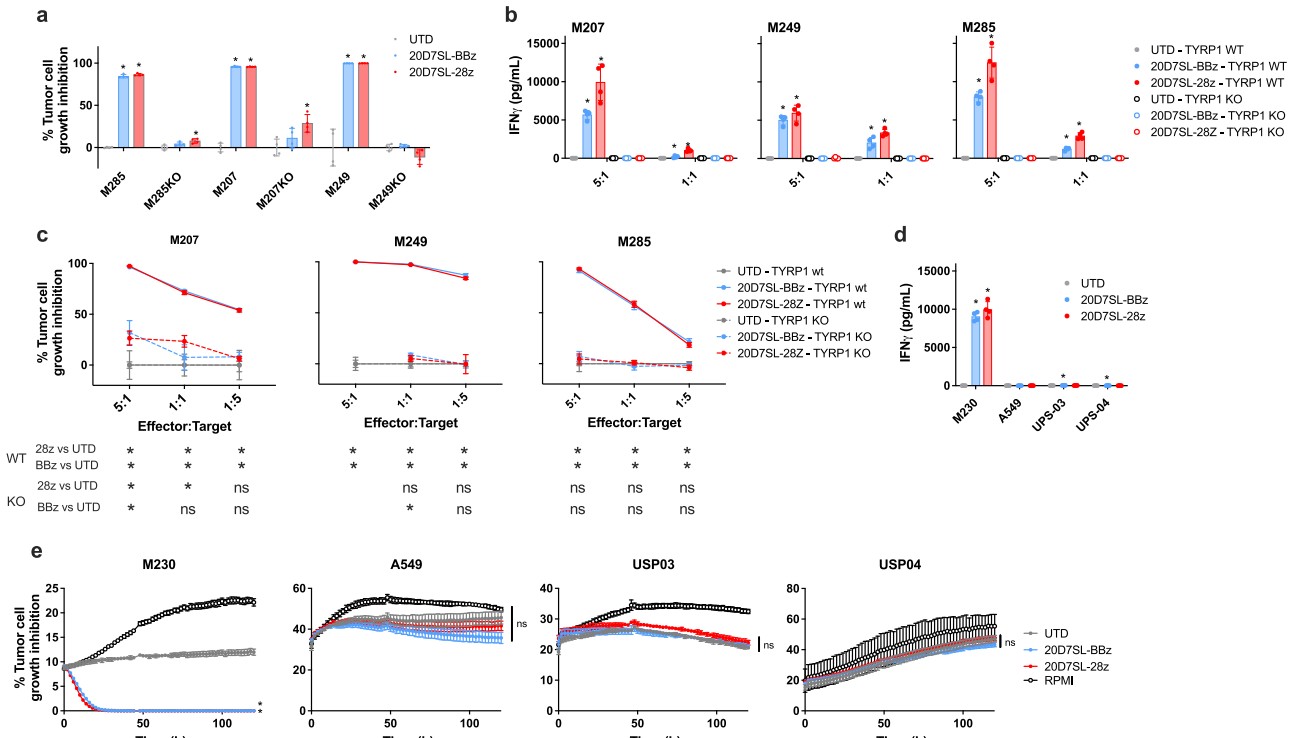

**Fig. 5 | 20D7SL-28ζ and 20D7SL-BBζ CAR-T cells specifically recognize TYRP1.**
**a**, **b** Co-culture of 20D7SL-28ζ and 20D7SL-BBζ with a panel of patient-derived TYRP1high cell lines and their *TYRP1*-knock out counterpart. Mean ± SD (*n* = 4, independent co-cultures) are plotted. *$p < 0.05$ vs untransduced T cells, unpaired, two-tailed *t* test with Holm-Sidak adjustment for multiple comparisons. **a** Percentage of tumor cell growth inhibition normalized by the growth of the cell lines co-cultured with untransduced T cells at 96 h after co-culture (1:1, E:T ratio). **b** IFNγ released at 24 h after co-culture measured by ELISA (5:1 and 1:1, E:T ratio). **c** Cytotoxicity dose–response curves of 20D7SL-28ζ and 20D7SL-BBζ CAR-T cells upon 48 h co-culture with a panel of TYRP1high melanoma cell lines and their *TYRP1*-knock out

counterparts. Percentage of tumor cell growth inhibition normalized by the untransduced T cells control. Mean ± SD (*n* = 4, independent co-cultures) are plotted. *$p < 0.05$, unpaired, two-tailed *t* test with Holm-Sidak adjustment for multiple comparisons. **d**, **e** Co-culture of 20D7SL-28ζ and 20D7SL-BBζ with a panel of a patient-derived TYRP1high cell line (M230) and non-melanoma cell lines. Mean ± SD (*n* = 4, independent co-cultures) are plotted. *$p < 0.05$ vs untransduced T cells, unpaired, two-tailed *t* test with Holm-Sidak adjustment for multiple comparisons. **d** IFNγ released at 24 h after co-culture measured by ELISA (5:1, E:T ratio). **e** Cytotoxicity time-course (5:1, E:T). Source data and exact *p* values are provided as a Source Data file.

between mice receiving untransduced or 20D7SL-28ζ CAR-T cells (Supplementary Fig. 6a–c). All changes in the blood cell counts were associated with irradiation, and no differences were observed between mice receiving untransduced or CAR-T cells. Interestingly, by day 13, we observed a significant decrease in aspartate aminotransferase and alanine aminotransferase levels and an increase in alkaline phosphatase levels in mice treated with 20D7SL-28ζ CAR-T cells compared to those treated with untransduced T cells or the untreated control group. The profile of these three parameters is associated with liver toxicity induced by tumor growth. Treatment with 20D7SL-28ζ CAR-T cells corrected the values for these enzymes to levels observed in tumor-free untreated mice and tumor-free mice receiving total body irradiation. These results highlight that the 20D7SL-28ζ CAR-T cell therapy not only controls tumor growth but also reverts the toxicity associated with the disease (Fig. 6h–j; Supplementary Fig. 6d–n). Total bilirubin and cholesterol were also decreased after treatment with 20D7SL-28ζ CAR-T cell therapy (Fig. 6k; Supplementary Fig. 6j, l). Finally, we quantified the proinflammatory cytokines released to the bloodstream on days 5, 13, and 27 after ACT. The levels of all cytokines (IL-6, IFNγ, IL10, IL12p70, MCP-1, and TNFα) remained below the detection level, and there was no peak associated with the treatment with 20D7SL-28ζ CAR-T cells (Fig. 6l–n; Supplementary Fig. 6o–q). We also measured long-term toxicities in immunocompetent mice without tumors receiving untransduced or 20D7SL-28ζ CAR-T cells. Similar to the results obtained on day 13, the 20D7SL-28ζ CAR-T cells were detected in the mice spleens at day 90, and no significant toxicities were observed based on body weight and condition, blood cell counts,

and chemistry (Supplementary Fig. 7). These results further support the lack of toxicity of the 20D7SL-28ζ CAR-T cells.

Of note, in some cases, C57BL/6 mice that completely cleared the tumor upon treatment with 20D7SL-28ζ CAR-T cells developed localized vitiligo where the tumors were implanted or in the back (Supplementary Fig. 8a). No other signs of toxicities attributable to targeting TYRP1 in normal pigmented cells, including the RPE, were observed. Specifically, we measured the retinal outer nuclei layer (ONL) thickness and the total TYRP1 expression in the RPE cells at days 13 and 27 after adoptive T-cell transfer with untransduced or 20D7SL-28ζ CAR-T cells and showed no differences between the treatment with 20D7SL-28ζ CAR-T cells or untransduced cells (Fig. 6o–q; Supplementary Fig. 8b–d).

### 20D7SL-28ζ CAR-T cell therapy is effective at treating acral and uveal melanoma
New therapeutic approaches for patients with rare melanoma subtypes are an unmet medical need. Given the high expression of *TYRP1* in biopsies from patients with acral, mucosal, and uveal melanoma, we tested the antitumor activity of the TYRP1 clinical candidate in models of these rare melanoma subtypes. First, we built panels of uveal and acral melanoma cell lines and compared the levels of *TYRP1* expression to cutaneous melanoma cell lines with high (M249) and intermediate (M202 and M229) *TYRP1* expression. In uveal melanoma, the five cell lines in our panel (MP65, MP38, MP41, MP46, MM28) presented high *TYRP1* expression (Fig. 7a). This high frequency of *TYRP1* overexpression in patient-derived cell lines is consistent with the levels

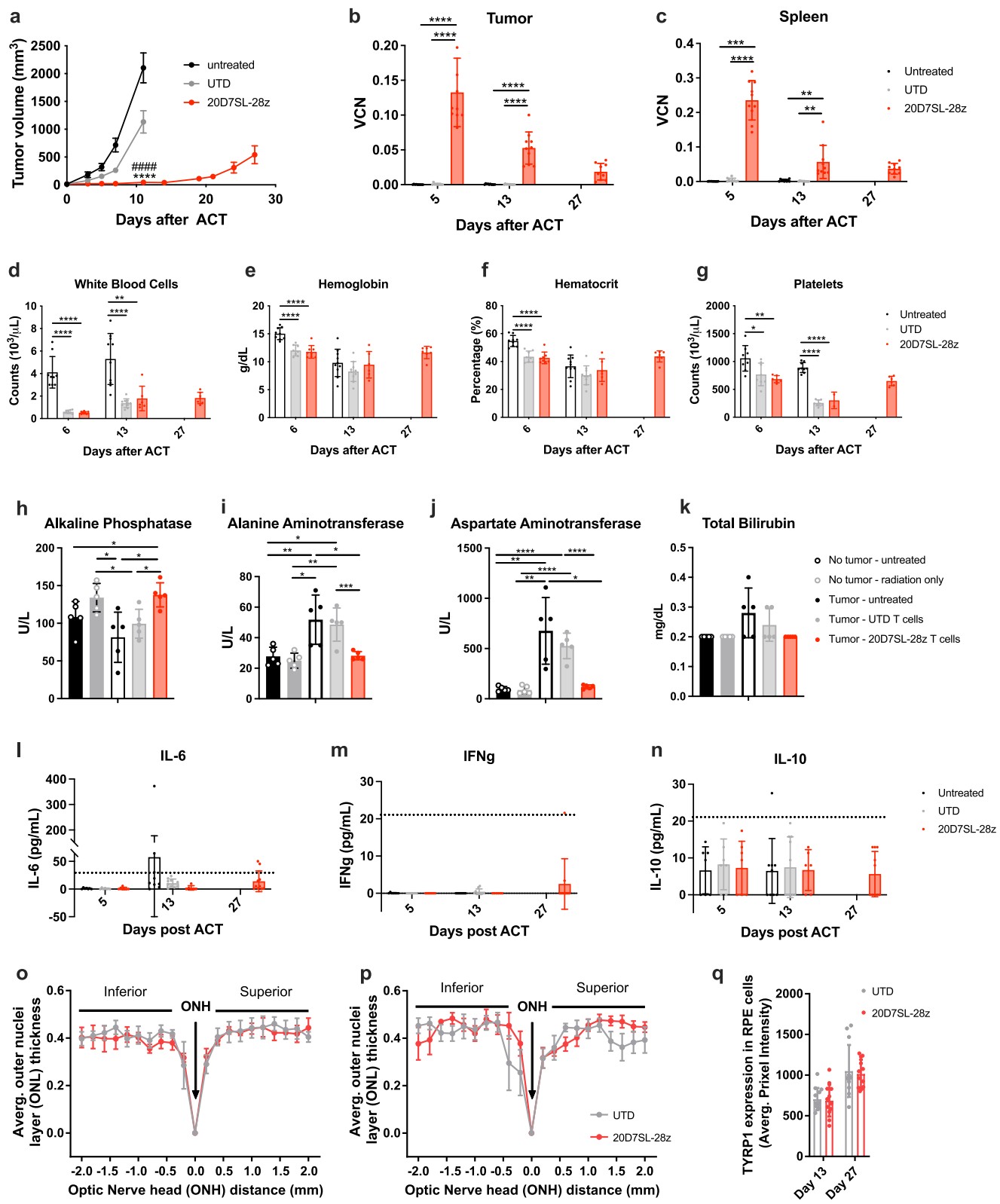

observed in tumor biopsies (Fig. 1f). All five cell lines were efficiently lysed by 20D7SL-28ζ CAR-T cells (Fig. 7b). Furthermore, co-culture with these five cell lines at different effector-to-target ratios led to 20D7SL-28ζ CAR-T cell activation, as measured by overexpression of 4-1BB in CD8+ T cells and OX-40 in CD4+ T cells expressing the CAR (Fig. 7c, Supplementary Fig. 11). Similarly, we established a panel of four acral melanoma cell lines (SK-Mel709B, SK-Mel1094A, SK-Mel990A, and SK-Mel1107A), two of which presented high *TYRP1*

expression (Fig. 7d), in line with the frequency of *TYRP1* over-expression observed in patient biopsies (Fig. 1d). Co-culture of 20D7SL-28ζ CAR-T cells with the two TYRP1high cell lines, SK-Mel709B and SK-Mel1094A, led to cytotoxicity as well as 4-1BB upregulation in CD8+ CAR-T cell and OX-40 upregulation in CD4+ CAR-T cells (Fig. 7e, f, Supplementary Fig. 11).

Finally, we measured the antitumor activity in vivo in immuno-deficient NSG mice bearing MP41 and MP46 uveal melanoma and

**Fig. 6 | 20D7SL-28ζ CAR-T cells exert strong tumor control and lack toxicity in the immunocompetent B16 melanoma model. a** Kinetic of tumor growth or regression over time after treatment with 20D7SL-28ζ CAR-T cells. Tumor size was measured with a caliper. Mean ± SEM is plotted ($n = 10$ mice for untreated, $n = 11$ mice for UTD and 20D7SL-28z). ****$p < 0.0005$ vs Untreated, ####$p < 0.0005$ vs UTD, unpaired, two-tailed $t$ test with Holm-Sidak adjustment for multiple comparisons. Retrovirus vector copy number in the tumor (**b**) and the spleen (**c**) of mice treated with 20D7SL-28ζ CAR-T cell therapy at days 5, 13, and 27 after adoptive T-cell transfer. Mean ± SD is plotted ($n = 10$ mice). Complete blood counts at days 5, 13, and 27 after adoptive T-cell transfer with 20D7SL-28ζ CAR-T cells. White blood cells (**d**), hemoglobin (**e**), hematocrit (**f**), and Platelets (**g**) are shown. Mean ± SD are plotted, each dot is one mouse (**d**–**f**) Untreated $n = 9$ for Day 6 and $n = 10$ for Day 13, UTD $n = 8$ for Day 6 and $n = 10$ for Day 13, 20D7SL-28z $n = 8$ for Day 6, $n = 6$ for Day 13 and $n = 7$ for Day 27; (**g**) Untreated $n = 8$ for Day 6 and $n = 9$ for Day 13, UTD $n = 8$ for Day 6 and $n = 9$ for Day 13, 20D7SL-28z $n = 6$ for Day 6, $n = 2$ for Day 13 and $n = 6$ for Day 27). Serum chemistry at day 13, after adoptive T-cell transfer with 20D7SL-28ζ CAR-T cells. Mice with tumors treated with untransduced T cells and vehicle were used as controls. Untreated mice without tumors and mice without tumors receiving irradiation only were used as additional controls. Alkaline phosphatase (**h**), Alanine aminotransferase (**i**), aspartate aminotransferase (**j**), and total bilirubin (**k**) are shown. Mean ± SD is plotted ($n = 5$ mice). Cytokine release in serum at days 5, 13, and 27 after adoptive T-cell transfer with 20D7SL-28ζ CAR-T cells. IL-6 (**l**), IFNγ (**m**), and IL-10 (**n**) are shown. Mean ± SD are plotted, each dot is one mouse (**l**–**n**) Untreated $n = 7$ for Day 5 and $n = 9$ for Day 13, UTD $n = 7$ for Day 6 and $n = 10$ for Day 13, 20D7SL-28z $n = 7$ for Day 6, $n = 6$ for Day 13 and $n = 10$ for Day 27). Black dashed line shows the minimum quantifiable levels using the BD CBA Mouse Inflammation Kit (20 pg/mL). **o, p** Quantitative measures of the average ONL thickness were taken at 0.2 mm intervals in frames (−10 to 10) to the optic nerve. Mean ± SEM is plotted ($n = 4$ mice). **q** Quantitative TYRP1 pixel intensity in the RPE cells from mice at day 13 and 27 after receiving ACT with untransduced T cells or 20D7SL-28ζ CAR-T cells. Mean ± SD is plotted ($n = 4$ mice, three sections of the retina per mouse). *$p < 0.05$, **$p < 0.005$, ***$p < 0.001$, ****$p < 0.0005$ unpaired, two-tailed $t$ test with Holm-Sidak adjustment for multiple comparisons. Unless otherwise indicated, differences are not statistically significant. Unless otherwise indicated, untransduced T cells and PBS were used as controls. Source data and exact $p$ values are provided as a Source Data file.

SK-Mel709B acral melanoma subcutaneous tumors. $3–5 \times 10^6$ CAR⁺-T cells were adoptively transferred, and untreated mice and mice receiving untransduced T cells were used as controls. Of note, the CAR-T cells for these experiments were manufactured using CD14⁻CD25⁻CD62L⁺ T naïve/stem cells and following the protocol that will be used in the phase I clinical trial. In all three tumor models, 20D7SL-28ζ CAR-T cells efficiently controlled tumor growth and led to 100% CR in both uveal melanoma models (Fig. 7g–i; Supplementary Fig. 9).

## Discussion

A major limitation to the success of CAR-T cell therapies for solid tumors is the scarcity of good tumor targets. The ideal CAR-T cell target is a protein consistently expressed on the surface of tumor cells but not on normal cells. Years of extensive research to identify potential targets for monoclonal antibody therapies have led to a few dozen candidates. Interest in these targets is now renewed with the emergence of CAR-T cell therapies and the ability not only to fine-tune the CAR structure design but also to express other modulatory proteins and improve the CAR-T cell manufacturing process to achieve highly functional and persistent T-cell responses. Using an iterative CAR optimization approach, we have developed a CAR-T cell therapy targeting TYRP1, a protein previously targeted clinically with a monoclonal antibody[26]. In our work, we demonstrate that targets used in monoclonal antibody therapy, with low-to-modest antitumor activity and lack of toxicity, can be repurposed and used to design more potent CAR-T cell therapies.

TYRP1 is an intracellular transmembrane protein that reaches the cell surface through the vesicular transport system and is constantly recycled from the melanosomes to the cell surface, and then endocytosed again. Only a fraction of the total TYRP1 is located on the cell surface at a given time-point; as a result, only cells with very high intracellular TYRP1 expression levels can present sufficient surface antigen to be detected by CAR-T cells. We took advantage of this dynamic range to increase the selectivity of CAR-T cells against TYRP1^high tumors while protecting against recognition of normal cells in the skin and the retina that endogenously express lower levels of TYRP1. Our clinical candidate, optimized through an iterative process, uses a clinically tested scFv (20D7S)[26], the CD28 transmembrane domain, and a long hinge (217 amino acids) derived from the immunoglobulin IgG4 that allows a high degree of flexibility to the CAR[41], and the CD28 costimulatory signal. The combination of these elements leads to robust in vitro and in vivo antitumor activity, lack of toxicity, and recovery from the toxicity induced by tumor growth. Of note, the TYRP1 CAR-T cell therapy exploits the vesicular transport system to target intracellular transmembrane proteins that localize on the cell membrane upon vesicle fusion. This approach opens an alternative avenue for identifying targetable surface proteins.

MDAs, such as MART-1 and gp100, have been previously used as targets for TCR-engineered T-cell therapy. An early clinical trial using gp100-specific T cells isolated from patients with metastatic melanoma expanded ex vivo and reinfused into the patients showed minimal efficacy but demonstrated the safety of the approach[42]. More recently, clinical trials using T cells genetically modified to express a MART-1 TCR have shown initial transient antitumor responses with cell-therapy–induced toxicities limited to cytokine release and depigmentation[39,43]. Most severe toxicities were attributed to the conditioning chemotherapy and high-dose IL-2 used to boost T-cell proliferation. Of note, a clinical trial using a high-affinity MART-1 TCR and a gp100 TCR showed toxicities in the skin, eye, and inner ear. These toxicities were transient and reverted with local steroid treatment[44]. Interestingly, in our previous clinical trial using the same high-affinity MART-1 TCR, we did not identify otologic, vestibular, or ophthalmologic toxicities after repeated specialist visits[39]. Studies in mice have shown that autoimmune destruction of the melanocytes in the eye is associated with local MHC up-regulation and that local prophylactic treatment with steroids does not impact the overall antitumor activity of the T cells[45]. Finally, it is relevant to highlight that the clinical trial investigating the 20D7SL monoclonal antibody[26] did not mention monitoring ocular and auditory function and had a protocol-required 30-day follow-up period. In this work, we propose to target TYRP1 with CAR-T cell therapy as a strategy to decrease the potential on-target off-tumor toxicity through selective action against TYRP1 over-expressing cells and increase the antitumor activity via antigen-binding–mediated primary and secondary T-cell signaling. Additionally, the lack of IL-2 supplementation requirement with the TYRP1 CAR-T cell therapy will further decrease the potential associated toxicities.

Based on the analysis performed on patient biopsies, we anticipate that ~30% of all melanoma patients without response or with relapse after ICB therapy will express high levels of TYRP1 and will be eligible for the TYRP1 CAR-T cell clinical trial. Based on the high TYRP1 expression in this patient subset and the antitumor activity observed in preclinical studies in immunocompetent mice and patient-derived models, we anticipate significant clinical responses and are preparing for the clinical translation. Interestingly, despite melanoma dedifferentiation being identified as a potential mechanism of resistance to cell therapies targeting MDA[46,47], the frequency of this phenomenon in treated patients is low[39,48].

Acral, mucosal, and uveal melanoma are rare melanoma subtypes with low incidence but poor prognosis due to the lack of response to

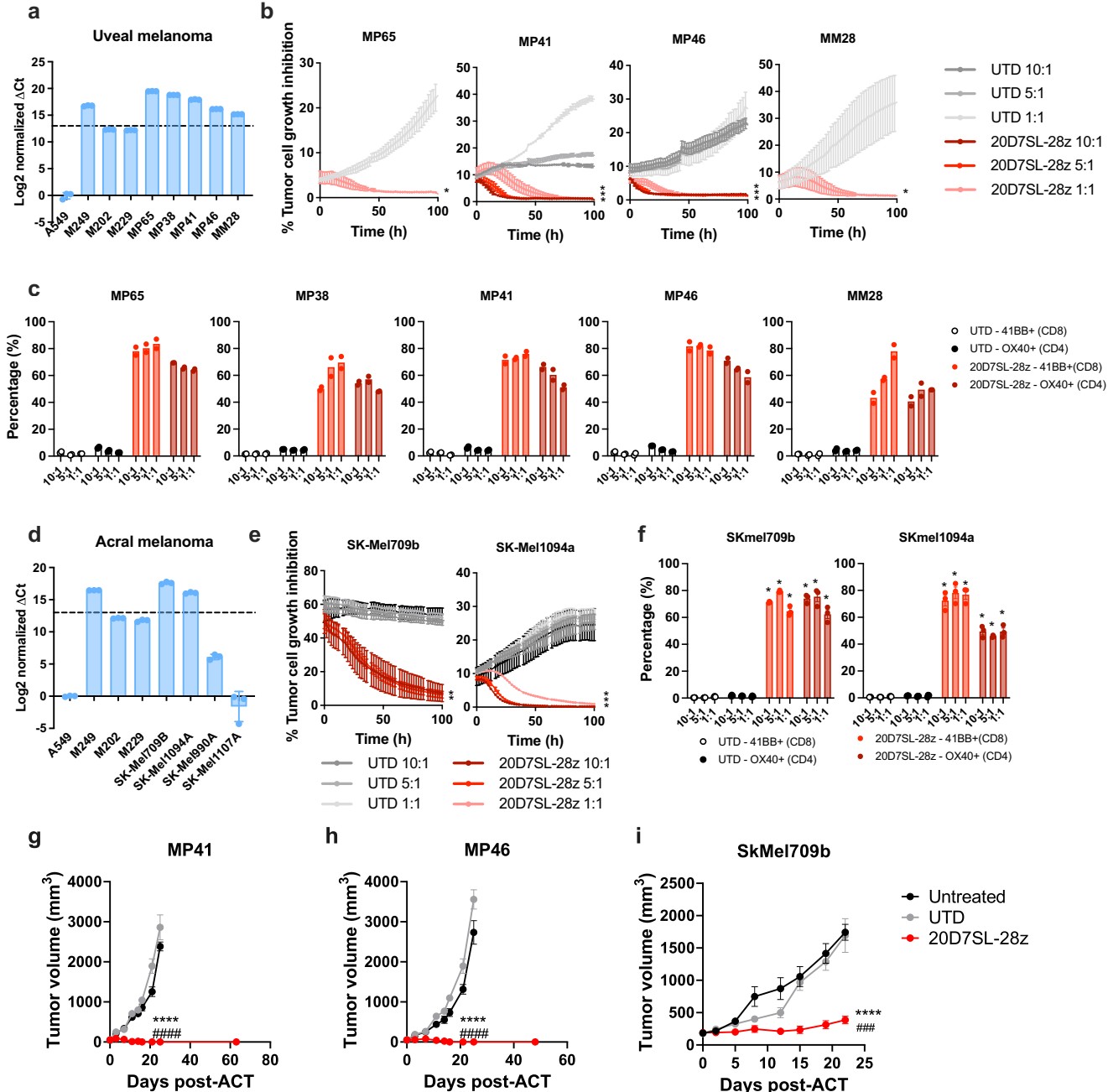

**Fig. 7 | 20D7SL-28ζ CAR-T cell therapy is effective at treating acral and uveal melanoma. a, d** *TYRP1* RNA expression measured by qPCR in a panel of uveal melanoma (**a**) and acral melanoma (**d**) cell lines. M249 TYRP1^high and M202 and M229 TYRP1^intermediate cutaneous melanoma cell lines were used as controls. A549 lung adenocarcinoma cell line was used as negative control. ΔCt is normalized by the Ct of the negative control cell line (*n* = 3 technical replicates, mean ± SD are plotted). **b, c** Antitumor activity of the 20D7SL-28ζ CAR-T cells upon co-culture with a panel of uveal melanoma cell lines at different E:T ratios (10:1, 5:1, and 1:1). **b** Time-course of uveal melanoma tumor cell growth and growth inhibition. Mean ± SD (*n* = 3 in all conditions except UTD 10:1 in MP41, which is *n* = 2, independent co-cultures) are plotted. **c** T-cell activation measured by the percentage of CD8⁺ T cells overexpressing 4-1BB and CD4⁺ T cells overexpressing OX-40. Mean (*n* = 2, independent co-cultures) is plotted. **e, f** Antitumor activity of the 20D7SL-28ζ CAR-T cells upon co-culture with a panel of acral melanoma cell lines at different E:T ratios. (10:1, 5:1, and 1:1) Mean ± SD are plotted (*n* = 3, independent co-cultures).

**e** Time-course of acral melanoma tumor cell growth and growth inhibition. **f** T-cell activation was measured by the percentage of CD8⁺ T cells overexpressing 4-1BB and CD4⁺ T cells overexpressing OX-40. *\*p < 0.05* vs untransduced, unpaired, two-tailed *t* test with Holm-Sidak adjustment for multiple comparisons. **g–i** Antitumor activity of the 20D7SL-28ζ CAR-T cells (derived from CD14⁻CD25⁻CD62L⁺ T cells) in vivo in patient-derived models of uveal and acral melanoma in immunodeficient mice. Kinetic of tumor growth or regression over time after treatment with 20D7SL-28ζ CAR-T cells. Tumor size was measured with a caliper. Mean ± SEM are plotted (*n* = 10 for all conditions except H untreated -*n* = 9- and I untreated -*n* = 8-)., *\*\*\*\*p < 0.0005* vs untreated, *### p < 0.001, #### p < 0.0005* vs untransduced T cells. Unpaired, two-tailed *t* test with Holm-Sidak adjustment for multiple comparisons. Unless otherwise indicated, untransduced T cells and PBS were used as controls. Source data and exact *p* values are provided as a Source Data file. Tables with titles and legends: NA.

standard melanoma treatments. Approximately 60% of patients with acral and mucosal melanoma and ~90% of patients with uveal melanoma overexpress TYRP1, and TYRP1 CAR-T cell therapy could potentially lead to clinical benefit in this patient subset. Importantly, tebentafusp, a bispecific fusion protein targeting gp100 presented by HLA-A*02:01, was recently approved by the FDA to treat metastatic uveal melanoma. Tebentafusp is a soluble high-affinity T-cell receptor binding domain specific for a gp100 peptide and fused to an anti-CD3 scFv[49]. With this design, tebentafusp recruits and activates T cells at the tumor site and induces the release of cytokines and cytolytic molecules against the tumor cells. Treatment with tebentafusp increases the overall survival and the progression-free survival as first-line therapy and in patients that relapsed or did not respond to previous treatments[50–52]. Remarkably, toxicities associated with tebentafusp treatment were mild to moderate and associated with cytokine release or skin toxicity, such as rush and pruritus, and did lead to treatment discontinuation in only 2% of the patients in the phase I trial[50–52]. A relevant limitation of tebentafusp is the HLA restriction, which limits its use to patients that express HLA-A*02. Based on the clinical success of tebentafusp and our preclinical results, we anticipate that the TYRP1 CAR-T cell therapy has the potential to benefit patients with uveal melanoma and overcome the HLA restriction, providing a clinical candidate for ~90% of patients with uveal melanoma.

## Methods

All research presented in this manuscript complies with all relevant ethical regulations under UCLA IRB approval 11–003254 and UCLA Animal Research Committee protocols # ARC-2004-159 and ARC-2021-040, which were previously approved by the Institutional Animal Care and Use Committee.

### RNA-seq analysis

For Figs. 1a, d–f, we combined three transcriptomic melanoma datasets: TCGA Research Network (http://cancergenome.nih.gov/), Abril-Rodriguez cohort[31] and Grasso cohort[30], to evaluate *TYRP1* log2 FPKMs expression values. For Fig. 1b, c, we used transcriptomic data from the study CheckMate 038 (NCT01621490) available in ref. 30. Raw counts were normalized to fragments per kilobase of exon per million fragments mapped (FPKM) expression values. Histogram and boxplots of log2 FPKM expression data were generated using the ggplot2 package in R software (version 3.5.1). For Fig. 1h, we used our melanoma cell line panel transcriptomic dataset[32] to evaluate *TYRP1* log2 FPKMs expression values.

### Immunohistochemistry

De-identified tissue from metastatic lesions of cutaneous, acral, mucosal, and uveal melanoma was obtained from the Pathology department at UCLA (collection and distribution under IRB 11-002504). Individual patient consent was not required for the release of de-identified remnant tissue, the UCLA Health Patient Procedure consent also states that leftover (remnant) biospecimens may be used for research purposes. Tissue was stained by the Translational Pathology Core Laboratory (TPCL) at UCLA. Briefly, 4 μm tissue sections from paraffin-embedded formalin-fixed tissue blocks were stained using a rabbit monoclonal primary antibody against TRP1 (Abcam, ab178676, Clone: EPR13063, cat# ab178676) at a 1:100 dilution for 1 h at room temperature. Rabbit IgG staining was used as negative control (Invitrogen Rabbit IgG isotype control, cat# 10500C). The staining was performed using a Leica BOND RX Automated Research Stainer (Leica Biosystems) with the Bond Polymer Refine Red Detection system. Routine factory-based "Bake and Dewax" protocol was followed with heat-induced antigen retrieval using ER2 buffer for 20 min. Prior to the primary antibody, BLOXALL™ blocking solution (LS Bio, LS-J1031) was used.

Mouse tumor tissue was extracted from subcutaneous tumors. Sample processing for TYRP1 and H&E staining was performed as described above. We used the Bond Prime Polymer DAB (3,3′-Diaminobenzidine) Detection system for TYRP1.

TYRP1 expression in tumor cell lines was also assessed by IHC. Briefly, cells were expanded until reached 30–60M cells. Cells were detached from the culture plates using 2 mM EDTA (ThermoFisher Scientific) and transferred to a 15 mL conical for future processing. Cells were washed with PBS and fixed o/n at 4 °C with 5 mL of GTF™ Formalin Substitute (StatLab). Next day, cells were washed three times with PBS and pelleted. Epredia™ HistoGel™ Specimen Processing Gel (Fisher Scientific) was used according to manufacturer's instructions to encapsulate and retain the cells before paraffin embedding. TYRP1 staining was performed on 4 μm sections as described above.

After staining, the slides were scanned in an APERIO brightfield scanner (Leica Biosystems) at a magnification of 40× and visualized using the Image Scope (v12.3.3.5048) and ImageJ (1.8.0_322) viewing software.

### TYRP1 expression

*TYRP1* mRNA expression was measured by RNAseq (as described in the RNAseq analysis section) or by reverse transcription PCR following the manufacturer's protocol for the Power SYBR Green RNA-to-CT 1-Step Kit (Applied Biosystems) and using the following primers: 5′-CAGGTTGTCTGGAGCAGTAAC-3′ and 5′- GGCTGAGGAGATACAA TGCTG-3′.

TYRP1 intracellular and surface protein expression was detected by flow cytometry. Briefly, tumor cells were seeded and grew for 24–48 h. For internalized staining, cells were treated with 2.5 μg of TYRP1 antibody (TA-99 clone, BioXCell or 20D7S/Flanvotumab, MyBioSource) and monensin (BD GolgiStop) to block internalized protein degradation for 16 h in culture, following the manufacturer's recommendations. Cells were detached from the culture plate using 2 mM EDTA (ThermoFisher Scientific) and transferred to V-bottom plates to perform the intracellular and surface stain. Cells were first stained with Zombie Violet (Biolegend) 1:100 dilution for 15 min at room temperature, and 50 μL of FBS was added to block the unspecific staining. For surface staining, 2.5 μg of TYRP1 antibody (TA-99 clone, BioXCell) or Mouse IgG2a, k Isotype Control were added and incubated for 30 min on ice. After washing, Anti Mouse IgG2A, k-APC was used as a secondary antibody. For intracellular and internalized staining, the cells were permeabilized using Fixation and Permeabilization kit (BD biosciences) according to manufacturers' instructions and stained with the same primary and secondary antibodies as used for surface stain. Flow cytometry acquisition was performed on an Attune NxT flow cytometer (Invitrogen).

### Construction of anti-TYRP1 CAR constructs

All anti-TYRP1 CAR constructs were derived from the 20D7S antibody[33]. All constructs were cloned into the splicing optimized pMSGV1 retroviral expression vector[37]. The pMSGV1-1G4 TCR vector was a kind gift from Dr. Steven Rosenberg (National Cancer Institute)[53]. The TYRP1-CARs were constructed by assembling the scFv (−n V$_L$ - V$_H$ orientation connected with the whitlow linker), the IgG4-derived hinge/spacer domains, the CD28 transmembrane and cytoplasmic domain, and the CD3ζ cytoplasmic domain. Following this approach, we generated the retroviral vector plasmids expressing the 20D7SS-28ζ, 20D7SM-28ζ, and the 20D7SL-28ζ CAR constructs. The difference between these three constructs is the hinge/spacer region. The 20D7SS-28ζ contains the IgG4 hinge sequence, 20D7SM-28ζ contains the IgG4 hinge and IgG4-CH3 loop, and the 20D7SL-28ζ contains the IgG4 hinge and the IgG4-CH2-CH3 loops. The IgG4-CH2 loop includes the L235E and N297Q mutations[36]. The pMSGV1-1G4 plasmid was digested using NcoI and BamHI, and the sequences of the 20D7S

antibody, hinges, transmembrane, and signaling domains were synthesized as gene block fragments (IDT Technologies). Vector and inserts were assembled using standard Gibson cloning strategies.

20D7SL-28ζ and the 20D7SL-BBζ CAR were also cloned in the epHIV7 backbone. The lentiviral backbone was a kind gift from Dr. Yvonne Chen (University of California, Los Angeles). The parental plasmid was digested with NheI and SmaI. The sequences for the CARs were amplified from previous versions of these plasmids, and the new sequences were synthesized as gene block fragments (IDT Technologies). Vector and inserts were assembled using standard Gibson cloning strategies.

All lentiviral and retroviral plasmids were sequenced from the 5′ LTR to the 3′LTR.

## Cell line generation and maintenance

Patient-derived cutaneous melanoma cell lines M202, M207, M229, M230, M249, M285, and M420 were established from patient biopsies with patient consent under UCLA IRB approval 11−003254, which has been approved by institutional policies. B16-F10 (murine melanoma cells, CRL-6475), A549 (human lung adenocarcinoma, CCL-185), and HEK-293T (Human embryonic kidney, CRL-1573) were purchased from ATCC. Undifferentiated pleomorphic sarcoma cell lines UPS-03 and UPS-04 were a kind gift from Dr. Anusha Kalbasi (University of California, Los Angeles). Uveal melanoma cell lines MP38, MP41, MP46, MP65, and MM28 were a kind gift from Dr. Chandrani Chattopadhyay (MD Anderson). Acral melanoma cell lines SK-Mel709B, SK-Mel990A, SK-Mel1094A, and SK-Mel1107A, were a kind gift from Dr. Taha Merghoub (Memorial Sloan Kettering Cancer Center, MSKCC).

All cutaneous melanoma cell lines, all undifferentiated pleomorphic sarcoma cell lines, A549, MP41, and MM28, were maintained in RPMI 1640 with L-glutamine (Thermo Fisher Scientific) containing 10% fetal bovine serum (FBS, Omega Scientific), penicillin (100 U ml$^{-1}$, Omega Scientific), streptomycin (100 μg ml$^{-1}$, Omega Scientific), amphotericin B (0.25 μg ml$^{-1}$, Omega Scientific), and Glutamax (Gibco). B16-F10 cell lines were maintained in Dulbecco's Modified Eagle Medium (DMEM, Corning) containing 10% FBS (Omega Scientific), penicillin, streptomycin, amphotericin B, and Glutamax. HEK-293T cells were maintained in DMEM containing 10% FBS (Hyclone, Cytiva), penicillin, streptomycin, amphotericin B, and Glutamax. MP46, MP38. and MP65 uveal melanoma cell lines were maintained in RPMI 1640 with L-glutamine containing 20% fetal bovine serum (Hyclone), penicillin, streptomycin, amphotericin B, Glutamax, and 1% ITS Liquid media supplement (1.0 mg/ml recombinant human insulin, 0.55 mg/ml human transferrin, and 0.5 μg/ml sodium selenite, Sigma). Acral melanoma cell lines were maintained in RPMI 1640 with L-glutamine containing 20% fetal bovine serum (Omega Scientific), penicillin, streptomycin, amphotericin B), and Glutamax. All cell lines were incubated at 37 °C in a humidified atmosphere of 5% $CO_2$. Cell lines were periodically authenticated using GenePrint 10 System (Promega) and were matched with the earliest passage cell lines. Selected melanoma cell lines were subjected to Mycoplasma tests periodically with the MycoAlert Mycoplasma Detection Kit (Lonza).

Cell lines used for cytotoxicity assays were stably transduced with a lentiviral vector expressing nuclear red fluorescent protein (nRFP) (NucLight Red Lentivirus EF1a Reagent, Essen Biosciences) and using protamine sulfate at 5 μg/mL as a transduction enhancer. A pure cell population was selected by cell sorting. The expression of nRFP facilitates cell growth measurement in the cytotoxicity and proliferation assays using real-time live cell imaging in an IncuCyte ZOOM (Essen Biosciences).

To generate the *TYRP1* knockout cell lines the following single guide RNA was used 5′- TCCCTGATGATGAGCCACAG-3′. Oligonucleotide forward: 5′- CACCGTCCCTGATGATGAGCCACAG-3′; reverse: 5′-AAACCTGTGGCTCATCATCAGGGAC-3′ were cloned into the pSpCas9(BB)-2A-GFP vector (Addgene) as described in Zhang's Lab

protocol[54]. Cells were then transfected with *TYRP1*-single guide RNA plasmid using lipofectamine 3000 (Thermo Fisher Scientific), and green fluorescent protein-positive cells were collected and single-cell sorted 48 h after transfection in 96-well plates at the UCLA Flow Cytometry Core. The clones were expanded. Genomic DNA was isolated for each clone (NucleoSpin Tissue XS; Macherey Nagel), a fragment of the *TYRP1* gene was amplified by PCR (primer forward: 5′-TACCAAGAAAAACTTGCATAATC-3′ and primer reverse 5′- CTAGTCATAATACCTAACACAGT-3′), and the PCR product sequenced using the same primers. The tracking of indels by decomposition[55] web tool was used to evaluate and confirm knockout efficiency (Supplementary Fig. 4b, d, e). TYRP1 deletion was also validated by western blot, performed as described previously[56]. TYRP1 antibody (Abcam, 1:10,000 dilution) and anti-rabbit IgG-HRP (Cell Signaling, 1:20,000 dilution) were used as primary and secondary antibodies, and blocking was done with PBS + 5% milk. Immunoreactivity was assessed with an ECL-Plus Kit (Thermo Scientific) and analyzed using the ChemiDoc MP system (Bio-Rad Laboratories) (Supplementary Fig. 4a, c).

## Retrovirus and Lentivirus production

Retrovirus supernatants were produced by transient transfection in HEK293T cells. The plasmids containing the retroviral vectors genome were co-transfected together with the retroviral vector packaging plasmids, pHIT60 and PRD114 (gifts from Dr. Yvonne Chen, University of California, Los Angeles) using TransIT293 (Mirus Bio). Sixteen hours after transfection, the cells were treated for 8 h with 10 mM Sodium Butyrate (Sigma) in the media. The retroviral vector supernatants were collected 40−48 h after transduction, filtered through a 0.45 μm filter, aliquoted, and stored at −80 °C. Lentiviral vectors were produced by transient transfection in HEK293T as described for the retroviral vector preparations but using the second-generation packaging plasmids pCMV-R8.9 and pVSV-G (a kind gift from Dr. Donald Kohn, University of California, Los Angeles). The vector supernatants were collected twice, at 40−44 h and at 66−70 h after transfection. The supernatants were filtered through a 45 μm filter and concentrated by ultra-centrifugation at 125,000 × g for 90 min at 4 °C. The virus pellets were resuspended, aliquoted, and stored at −80 °C. The vectors from the two collections were kept separated. The functional titer of the lentiviral vector preparations was measured in HT29 cells by the Kohn Preclinical Lab at UCLA.

## Human CAR-T cell generation

To generate CAR-T cells, total PBMCs were obtained from healthy donors consented under IRB 10-001598, purified by Ficoll-Paque PLUS (GE Healthcare) density gradient, and cryopreserved. A day before starting the CAR-T cell production, PBMCs were thawed at 37 °C and incubated overnight with RPMI 1640 with L-glutamine containing 10% FBS (Hyclone), penicillin, streptomycin, amphotericin B, Glutamax at 37 °C and 5% $CO_2$. The next day, PBMCs were activated with a 1:1 ratio of CD3:CD28 beads (Invitrogen) and 500 IU/mL of hIL2 (Miltenyi Biotec). Forty to 48 h after activation, T cells were transduced with retroviral vector supernatants in retronectin-coated plates. Briefly, 4 mL of retrovirus supernatant were spinoculated in retronectin-coated six-well plates for 2 h at 2000 × g with no brake. When centrifugation ended, 2 mL of the supernatant were removed and 4 mL of activated T cells at a concentration of 0.375−0.5 M/mL were added to each well. Cells were spinoculated for 10 min at 2000 × g no break. After transduction, cells were expanded in RPMI 1640 with L-glutamine containing 10% FBS (Hyclone), penicillin, streptomycin, amphotericin B, Glutamax, and 300 IU/mL hIL2. At day 4−6 after transduction, the transduction efficiency was measured by flow cytometry using antibodies to detect the whitlow linker (Kip-1 antibody, a kind gift from Kite Pharma) or F(ab′)2 anti-human IgG (H + L) (Jackson Immunoresearch Laboratories, cat# 109-005-006). When using lentiviral vectors, CAR-T cells were manufactured using a similar protocol as

described above, but cells were transduced in tissue-culture treated plates with 0.5 TU/cell of virus and using 0.25 mg/mL of LentiBoost (Sirion Biotech) as a transduction enhancer. For the experiments in acral and uveal melanoma models, CAR-T cells were manufactured using a subset of CD14⁻CD25⁻CD62L⁺ naïve/memory T cells as described in ref. 57. Briefly, the CD14⁻CD25⁻CD62L⁺ population was isolated by an initial depletion of CD14+ and CD25+ cells using CD14 and CD25 microbeads (Miltenyi Biotec) and LD columns (Miltenyi Biotec) and the MidiMACs magnet (Miltenyi Biotec) and a consecutive CD62L enrichment step using CD62L microbeads (Miltenyi Biotec) and LS columns (Miltenyi Biotec) and the MidiMACs magnet. After verifying the purity of the population by flow cytometry using CD14 (Miltenyi Biotec, mouse IgG2ak, clone TUK4, cat# 130-113-145), CD25 (Miltenyi Biotec, mouse IgG2bk, clone 4E3, cat# 130-113-280), and CD62L (Miltenyi Biotec, mouse IgG1k, clone 145/153, cat# 130-113-619) antibodies, the cells were aliquoted and cryopreserved for future use. This subset of T cells was activated and transduced using lentiviral vectors as described above to manufacture TYRP1 CAR-T cells.

### T-cell functional studies: T-cell activation, cytokine release, cytotoxicity, and repeated antigen challenges

We measured T-cell activation, cytokine release, and cytotoxicity upon co-culture of the anti-TYRP1 CAR-T cells and several melanoma (M202, M207, M229, M230, M249, M285, M207-TYRP$^{KO}$, M230-TYRP$^{KO}$, M285-TYRP$^{KO}$, MP38, MP41, MP46, MP65, MM28, SK-Mel709b, SK-Mel1094a, and B16) and non-melanoma cell lines (A549, UPS-03, UPS-04). Briefly, for co-cultures, target tumor cells were seeded in 96-well plates and incubated overnight. $25 \times 10^3$ cells/well were plated for cytokine secretion and cytotoxicity studies, and $10 \times 10^3$ cells/well (except $40 \times 10^3$ cell/well for SkMel709b) were plated for the T-cell activation studies. For experiments using B16 $5 \times 10^3$ cells/well were seeded. The next day, untransduced or CAR-T cells were added following the effector-to-target ratio as indicated in each figure. To measure cytokine release, we collected the co-culture supernatant 20–24 h after co-culture and stored it at −80 °C. When ready for the analysis, the supernatants were thawed on ice, and the concentration of IFNγ was measured by ELISA (ThermoFisherScientific).

To measure T-cell activation, after 20–24 h co-culture, T cells were collected, washed with PBS, and stained with Zombie Violet live/dead stain (1:100, Biolegend) in PBS. After incubation, cells were stained with CD45 clone HI30 (FITC, Biolegend, cat# 304005 or PE, BD), CD4 clone OKT4 (BV421 or BV510, Biolegend, cat# 317433 and cat# 317443, respectively), CD8 clone RPA-T8 (BV605, Biolegend, cat# 301039), 41BB clone 4B4-1 (APC, Biolegend, cat# 309809) and OX40 clone Ber-ACT35 (PE-Cy7, Biolegend, cat# 350012) at 1.25 μL of each antibody per sample.

In all flow cytometry staining experiments, after the last incubation with antibodies, cells were then washed, fixed, and stored at 4 °C until flow cytometry acquisition. All stains and washes were performed in PBS unless otherwise indicated. Flow cytometry acquisition was performed on an Attune NxT flow cytometer (Invitrogen) following the gating strategy shown in Supplementary Fig. 11.

To measure cytotoxicity, nRFP⁺ cell lines were used. These cell lines were generated as described in the "cell line generation and maintenance" section. $25 \times 10^3$ cells/well of the nRFP⁺ cell lines were seeded in 96-well plates and incubated overnight. The next day, CAR-T cells were added following the effector-to-target ratio as indicated in each experiment. After adding the T cells, melanoma cells were imaged using a real-time live cell imaging system (Incucyte, Essen Biosciences) as a time course or at the indicated time points.

For repetitive antigen challenge experiments, $30 \times 10^3$ cells/well of M230nRFP and M249nRFP cells were seeded and incubated overnight. The next day, CAR-T cells were added using a 1:1 effector:target ratio and incubated for 48 h. Every 48 or 72 h, $30 \times 10^3$ cells/well of M230nRFP or M249nRFP were added to each well. The melanoma cell growth and killing were followed over time and imaged using a real-time live cell imaging system (Incucyte, Essen Biosciences).

Untransduced T cells and media alone were used as a control in all experiments. All functional studies were done in melanoma cell media, not supplemented with cytokines. Cytotoxicity experiments were done in biological quadruplicates unless otherwise stated.

### In vivo studies
All animal experiments were performed under the UCLA Animal Research Committee protocols # ARC-2004-159 and ARC-2021-040, which were previously approved by the Institutional Animal Care and Use Committee. Six- to eight-week-old C57BL/6N and NOD/SCID/IL-2Rgnull (NSG) mice were obtained from the UCLA Radiation Oncology Animal facility colony from the Division of Laboratory Animal Medicine and housed in the same facility. Mice were housed in individually ventilated, autoclaved standard cages and bedding supplied with HEPA filtered air. Males and females were housed separately. Unless otherwise stated, female mice were used for all in vivo experiments. Mice were housed in individually ventilated racks. The rack air is HEPA-filtered via the rack blowers. Room temperature is maintained between 20 and 26 °C (68–79 °F) monitored and recorded daily. Room humidity is maintained between 30 and 70% and is also monitored and recorded daily. The test facility is an AAALAC-accredited facility and meets all air quality standards stated in the Guide for the Care and Use of Laboratory Animals (Guide), Eighth Edition (National Research Council 2011).

For in vivo studies in immunocompetent C57BL/6 mice, B16 murine melanoma cells were cultured in DMEM w/L-glutamine medium (Corning) supplemented with 10% heat-inactivated FBS (Omega Scientific), 1X PSA (Omega Scientific) and 2 mM Glutamax (Gibco) and maintained in a humidified incubator (37 °C and 5% CO₂). On Day −5, B16 cells ($0.5–0.3 \times 10^6$ cells/tumor) were prepared in a final volume of 100 μL PBS and implanted in the right flank of the leg in 6–10 weeks old female C57BL/6 mice. UCLA Ethics Committee does not judge tumor burden on a generic size criterion, since the impact of a tumor will vary depending on its location. For consistency, in all data presented in this manuscript mice were euthanized when their tumor reached a maximum volume of 2500 mm³ or earlier if deemed necessary for health reasons. In some cases, this limit was exceeded on the last day of measurement, and the mice were immediately euthanized.

For adoptive cell transfer (ACT), T cells were isolated from spleens from C57BL/6 mice (Day −5 or −4), activated with CD3/28 beads (Gibco) using a 1:1 ratio and mIL-2 (Peprotech) at 40 IU/mL and incubated for one day in RPMI media (Corning) supplemented with 10% FBS (Hyclone), 0.05 mM 2-mercaptoethanol (Gibco), 1X PSA, 20 mM HEPES (Gibco), 1 mM sodium pyruvate (Gibco). The next day (Day −4 or −3), T cells were collected and transduced with the retroviral vectors expressing the anti-TYRP1 CAR constructs (20D7SS-28ζ, 20D7SM-28ζ, and/or 20D7SL-28ζ) by spinoculation in retronectin coated plates using a 1:1 mix of vector supernatant and fresh media. After transduction, cells were expanded for 4–3 more days. Untransduced T cells were generated in parallel to be used for the control groups and treated exactly as the transduced T cells.

T cells were administered (Day 0) intravenously via tail vein injection to the C57BL/6 mice that had received body irradiation with 400–500 cGy the day prior to cell administration (Day −1). Before injection, transduction efficiency was analyzed by flow cytometry using the Kip-1-PE (kind gift from Kite Pharma) or Fab2-Alexa Fluor 488 (Jackson Immuno Research) to calculate the number of CAR⁺ cells to administer ($2.8–7 \times 10^6$ CAR+ cells). Tumor growth was measured three times a week using a caliper. Tumor volume was calculated using the formula $(L \times w^2)/2$, where $L$ is the longest diameter and $w$ is the shortest diameter of the tumor. For survival studies, mice were euthanized when their tumor reach 2500 mm³.

For the toxicology studies, after ACT, tumor growth, mass, and temperature were monitored 2–3 times/week. Mice were euthanized at day 5, 13, or 27 after ACT, and hematology, serum chemistry, serum cytokine quantification, and retrovirus vector copy number (VCN) in the spleen and tumor were performed. Blood samples for hematology and serum chemistry were processed by Quality Vet Lab (Davis, CA). Blood smears were also performed as a backup. Between 50–100 µL of serum were collected from the serum separator tube, for cytokine quantification. Cytokine quantification was assessed using the BD™ CBA Mouse Inflammation Kit (BD Biosciences) following the manufacturer's instructions. The minimum and maximum quantifiable levels using the BD CBA Mouse Inflammation Kit are 20–5000 pg/mL. Due to the reduced sensitivity of the Mouse MCP-1 assay, we established 80pg/mL as the minimum level of detection for MCP-1. Spleens and tumors were processed to obtain single-cell suspensions. Briefly, spleens single-cell suspension was obtained by dissociating the spleen on the mesh strainer (70 µm) with a syringe plunger, additionally the red blood cells were lysed by incubation with ACK lysis buffer. Tumors were dissociated using Tumor Dissociation kit (Miltenyi Biotec, murine) and the gentleMACS dissociator (Miltenyi Biotec) following the manufacturer's instructions. Cell pellets of $2 \times 10^6$ cells were stored and used for DNA isolation using PureLink Genomic DNA mini Kit (Invitrogen). These DNAs were used for VCN determination.

Designated mice were used for the RPE toxicity studies. Light microscopy analysis of plastic-embedded mouse eyes was performed as previously described[58]. Briefly, mice were euthanized and the superior pole of the mouse eyes was marked with a cautery pen for orientation before enucleation. The eyes were fixed in 2% formaldehyde and 2.5% glutaraldehyde in 0.1 M sodium phosphate buffer and cut into temporal and nasal hemispheres. Fixed hemispheres were immersed for 1 h in 1% osmium tetroxide dissolved in 0.1 m sodium phosphate buffer (pH 7.4) followed by dehydration in a graded series of alcohols and embedded in an Epon/Araldite mixture. Retina sections were cut at 1 µm, stained with 1% toluidine blue and 1% sodium borate, and then photographed with a Zeiss Axiophot microscope fitted with a 40× oil immersion objective and a CoolSNAP digital camera. Serial images (5–6 frames) of the superior and inferior optic nerve head were taken and stitched up subsequently using the NIH Image J software[59]. The thickness of the ONL corresponding to the photoreceptor cells was measured in 200 µm intervals from the optic nerve head.

TYRP1 immunohistochemistry of frozen mouse eye was performed as previously described[60], mice were euthanized and the superior pole of the mouse eyes was marked with a cautery pen for orientation before enucleation. Enucleated eyes were fixed in 4% formaldehyde with 0.1 M sodium phosphate buffer (NaPO4, pH 7.4) overnight. After rinsing off the fixative, the eyes were dissected in 0.1 M NaPO4, and the anterior segment (cornea, lens, vitreous) was removed to obtain the RPE-choroid-scleral eyecups. The eyecups were infiltrated with 10–30% sucrose for cryoprotection, embedded in Optimal Cutting Temperature embedding medium (OCT; Tissue-Tek), and 10-µm frozen retina sections were cut for staining experiments. After washing in 3× with phosphate buffer, frozen sections were quenched with 50 mM NH4Cl, and blocked with 5% goat or donkey serum with 1% BSA in PBS. The sections were then exposed to rabbit polyclonal antibody to TYRP1 (1:100 Biorbyt) overnight at 4 °C, washed with phosphate buffer, and incubated with the secondary antibody-conjugated Alexa Fluor dye for 1 h at room temperature. All sections were stained with DAPI nuclear marker (Invitrogen) and mounted with 5% n-propyl gallate in 100% glycerol. Images were captured using an Olympus FV 1000 confocal microscope (60× objective).

For in vivo studies in immunodeficient NSG mice, patient-derived human melanoma cell lines ($5 \times 10^6$ M207 cells, $1 \times 10^6$ M249 cells, $1 \times 10^6$ M230 cells, $2 \times 10^6$ MP41 cells, $2 \times 10^6$ MP46 cells, $2 \times 10^6$ SK-Mel709B cells) were implanted subcutaneously. Melanoma cells were prepared in a final volume of 100 µl PBS or Matrigel:RPMI (1:1, Corning) and injected subcutaneously in 6–10 weeks-old female NSG mice. Two tumors per mouse were implanted in the flanks. When the tumors engrafted and started growing (7–41 days), CAR⁺ T cells were injected systemically via the tail vein. Before injection, transduction efficiency was analyzed by flow cytometry using the Kip-1-PE (kind gift from Kite Pharma) or Fab2-Alexa Fluor 488 (Jackson Immuno Research) to calculate the number of CAR⁺ cells to administer/mice–($3.3 - 6 \times 10^6$ CAR⁺ cells/mice). CAR-T cells were manufactured as indicated above. For experiments with cutaneous melanoma models CAR-T cells were manufactured using the total PBMC population, for acral and uveal melanoma models T^naive/memory (CD14⁻CD25⁻CD62L⁺) subset was used. Tumor growth was measured 2–3 times a week using a caliper. Mice were euthanized when tumors in each group reached 2500 mm³ or signs of GVHD appeared unless otherwise indicated.

For in vivo flow cytometry analysis, tumors were digested with 50 u/ml DNase (Sigma) and 1mg/ml Collagenase (Sigma) for 1 h at 37 °C. Spleens were dissected manually. Single-cell suspensions were obtained using 70 µm strainers. $1 \times 10^6$ cells were stained with Zombie NIR (Biolegend) 1:100 for 15 min at room temperature. After a wash, cells were blocked with anti-mouse CD16/32 (Invitrogen) and stained with anti-mouse CD3-eFluor450 (eBiosciences, Clone SK7, cat# 48-0036-42) or anti-human CD45-FITC (Biolegend, clone HI30, cat# 304005) for 30 min at 4 °C. Flow cytometry acquisition was performed on an Attune NxT flow cytometer (Invitrogen). The gating strategy is shown in Supplementary Fig. 10.

## Vector Copy Number

VCN was determined for all DNAs via digital droplet PCR (ddPCR; BioRad), as described in ref. 61, and using the primer/probe sets (Integrated DNA Technology) specific for the retroviral vectors (for murine CAR-T cell assessment) or specific for the lentiviral vector (for human CAR-T cell assessment). A primer/probe set for an internal housekeeping gene was used as an internal calibrator. We used the mouse-specific uc378 region for murine T cells and the syndecan-4 gene for human T cells.

The primers and probes used for murine CAR-T cell VCN assessment were the following: for the retroviral vector, MSGV1-psi-ddFP1 (GCCTGTTACCACTCCCTTAAGT), MSGV1-psi-ddRP1 (GGCCATCCGACGTTAAAGGT) and MSGV1-psi-probe (/56-FAM/TCGGTAGAT/ZEN/GTCAAGAAGAGACGTTGGGTT/3IABkFQ/); for the internal calibrator, uc378 F (CGCCCCCTCCTCACCATTAT), uc378 R (CATCACAACCATCGCTGCCT), uc378 HEX probe (/5HEX/TTACCTTGC/ZEN/TTGTCGGACCAAGGCA/3IABkFQ/). The primers and probes used for human CAR-T cell VCN assessment were the following: for the lentiviral vector, HIVU5 F (AAGTAGTGTGTGCCCGTCTG), HIVpsi R (CCTCTGGTTTCCCTTTCGCT) and HIV1U5 probe (/56-FAM/CCCTCAGAC/ZEN/CCTTTTAGTCAGTGTGGAAAATCTCTAG/3IABkFQ/); for the internal calibrator, SDC4 FWD (CAGGGTCTGGGAGCCAAGT), SDC4 REV (GCACAGTGCTGGACATTGACA), SDC4 HEX probe (/5HEX/CCCACCGAACCCAAGAAACTAGAGGAGAAT/3IABkFQ/).

## Statistics and reproducibility

In CAR-T cell cytotoxicity, cytokine release, and T-cell activation studies, a two-sided unpaired $t$-test was used to compare the different CAR constructs and the untransduced T cells or media alone used as negative controls. Holm-Sidak adjustment for multiple comparisons was used where more than one group was compared, as indicated in each figure legend. For in vivo antitumor efficacy studies two-sided unpaired $t$ test with Holm-Sidak adjustment for multiple comparisons was used to assess the significance of the activity between the TYRP1 CAR-T cell treated groups and the negative controls. Irrelevant CD19 CAR-T cells, untransduced T cells, or vehicle were used as negative controls. Log-rank (Mantel-Cox) was used to determine the statistical significance of the differences in mice survival. A two-sided unpaired $t$

test was used to compare differences in persistence of the CAR-T cells and the toxicological parameters in vivo. Holm-Sidak adjustment for multiple comparisons was used where more than one group was compared, as indicated in each figure legend. No statistical method was used to predetermine sample size. No data were excluded from the analyses. In vivo studies were randomized and mice were divided between groups to equalize the starting tumor volume. The rest of the experiments were not randomized. The Investigators were not blinded to allocation during experiments and outcome assessment.

The number of biological replicates and significance are indicated in each figure legend. Exact *p* values are shown in the Source data file.

### Reporting summary
Further information on research design is available in the Nature Portfolio Reporting Summary linked to this article.

## Data availability
All sequencing data used had been previously published and can be bound in TCGA Research Network (http://cancergenome.nih.gov/), the Abril-Rodriguez cohort (phs001919)[31], the Grasso cohort (EGAS00001004545)[30], and the Ribas Laboratory melanoma cell line panel transcriptomic dataset (GSE80829) published in ref. [32]. Graphical depictions were created with BioRender.com. Source data are provided in this paper.Further information and requests for resources and reagents should be directed to Cristina Puig-Saus (cpuigsaus@mednet.ucla.edu). Source data are provided with this paper.

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

## Acknowledgements

This study was funded in part by the Parker Institute for Cancer Immunotherapy (PICI, funding to A.R. and C.P.-S.) and by a DoD CDMRP-TRA grant (W81XWH2110839, grant to C.P.-S), a grant from the California Institute for Regenerative Medicine (CIRM-TRAN1 12258, grant to C.P.-S), a Young Investigator Award from the Melanoma Research Alliance (MRA, 824855, grant to C.P-S), and the Ablon Scholar Award (to C.P-S). This study was also funded by NIH grants R35 CA197633 (grant to A.R.) and P01 CA244118 (grant to A.R.), the Ressler Family Fund, and generous contributions from Ken and Donna Schultz, Todd and Donna Jones, and Thomas Stutz (to A.R.). C.P.-S. is a Parker Senior Fellow supported by PICI. Ocular investigations were supported by the P30-EY000331 Stein Eye Institute Core Grant for Vision Research (R.A.R.). We acknowledge Chunni Zhu for their technical assistance with retinal microscopy studies. G.A.-R. was supported by the Isabel and Harvey Kibel Fellowship award and the Alan Ghitis Fellowship Award for Melanoma Research, and currently, by a Parker Scholar award. Flow cytometry was performed in the UCLA Jonsson Comprehensive Cancer Center (JCCC) Flow Cytometry Shared Resource that is supported by the National Institutes of Health award P30CA016042 and by the JCCC and the David Geffen School of Medicine at UCLA. Cell sorting was performed at the "Eli and Edythe Broad Center of Regenerative Medicine and Stem Cell Research University of California, Los Angeles Flow Cytometry Core Resource. We also acknowledge the Technology Center for Genomics and Bioinformatics, the Division of Laboratory Animal Medicine (DLAM), and Translational Pathology Core Laboratories (TPCL) at UCLA.

## Author contributions

C.P.-S., A.R., and Y.Y.C. designed the study and provided overall guidance. C.P.-S wrote the first version of the manuscript. All authors contributed to the final manuscript. C.P.-S., E.M., G.A.-R., A.P.-M. analyzed the results. C.P.-S. and G.A.-R. contributed to bioinformatics analysis. C.P.-S., E.M., S.J., J.D.S., D.B.-M. contributed to TYRP1 CAR-T cell development and antitumor efficacy studies, C.P.-S., J.C., C.N., A.P.-M., A.P., J.H., and R.A.R. contributed TYRP1 CAR-T cell safety studies. C.P.-S., R.G. contributed to CAR-T cell manufacturing studies. C.P.-S., A.P.-M., D.S., and P.S. contributed to patient sample collection, processing, and development CLIA-certified assay to detect TYRP1. A.V.-C., I.B.-C., and J.M.C. contributed to regulatory issues and logistics.

## Competing interests

A.R. has received honoraria from consulting with Amgen, Bristol-Myers Squibb, Chugai, Dynavax, Genentech, Merck, Nektar, Novartis, Roche, and Sanofi, is or has been a member of the scientific advisory board and holds stock in Advaxis, Arcus Biosciences, Bioncotech Therapeutics, Compugen, CytomX, Five Prime, RAPT, ImaginAb, Isoplexis, Kite-Gilead, Lutris Pharma, Merus, PACT Pharma, Rgenix and Tango Therapeutics. A.R. is a co-founder of PACT Pharma, a member of the Board of Directors, and holds founder stock. Y.Y.C., A.R., and C.P.-S. are founders of, hold equity in, are members of the Scientific Advisory Board, and receive consulting fees from ImmPACT Bio. Y.Y.C. is a member of the scientific advisory board and holds equity in Catamaran Bio, Notch Therapeutics, Pluto Immunotherapeutics, Prime Medicine, Sonoma Biotherapeutics, and Waypoint Bio. She has consulted for Novartis and Gritstone Bio.

UCLA has filed Patent applications on aspects of the described work, entitled "Chimeric Antigen Receptors and Related Methods and Compositions for the Treatment of Cancer" (WO2021046432A1). No potential competing interests were disclosed by the other authors.
