## [Peer Review File · Nature Communications]

REVIEWERS' COMMENTS

Reviewer #1 (Remarks to the Author):

The manuscript "CAR-T cell therapy targeting TYRP1 surface expression to treat cutaneous, acral, mucosal, and uveal melanoma" by Jilani, Saco, Mugarza et al., describes targeting an intracellular melanosomal membrane protein, TYRP1 (also known as TRP-1) with a chimeric antigen receptor (CAR). This is possible because the protein cycles from the melanosome to the outer cell membrane via vesicular transport. Function of their affinity-selected, meticulously constructed second generation receptor is well documented in vitro and in murine models. They emphasize the applicability of this target to acral, mucosal and uveal melanomas which respond poorly to other immunotherapies due to low mutational burdens. Like other T-cell therapies targeting melanosomal proteins, this one may have issues with vitiligo and possible eye and ear toxicities although the authors could not detect the latter pre-clinically. As the planned clinical trial will not administer concomitant IL-2, it will be interesting to see if IL-2 had a role in MHC upregulation and the normal tissue toxicity of prior MHC-dependent receptor protocols. The authors have also addressed the potential for antigen-loss variants as an escape mechanism. Their data does little to assuage that possibility for some tumors, but clearly others do not demonstrate the phenomenon.

The authors have presented thorough and forthright responses to the issues raised in the initial review and I have no further questions.

Reviewer #3 (Remarks to the Author):

I believe the conclusions of the manuscript are well supported and the authors have answered all of the questions raised by the original reviewers adequately.

I have only one minor comment that I personally found the title on line 281 confusing as I was not sure what panel was being referred to.

Reviewer #1 (Remarks to the Author):

The manuscript “CAR-T cell therapy targeting TYRP1 surface expression to treat cutaneous, acral, mucosal, and uveal melanoma” by Jilani, Saco, Mugarza et al., describes targeting an intracellular melanosomal membrane protein, TYRP1 (also known as TRP-1) with a chimeric antigen receptor (CAR). This is possible because the protein cycles from the melanosome to the outer cell membrane via vesicular transport. Function of their affinity-selected, meticulously constructed second generation receptor is well documented in vitro and in murine models. They emphasize the applicability of this target to acral, mucosal and uveal melanomas which respond poorly to other immunotherapies due to low mutational burdens. Like other T-cell therapies targeting melanosomal proteins, this one may have issues with vitiligo and possible eye and ear toxicities although the authors could not detect the latter pre-clinically. As the planned clinical trial will not administer concomitant IL-2, it will be interesting to see if IL-2 had a role in MHC upregulation and the normal tissue toxicity of prior MHC-dependent receptor protocols. The authors have also addressed the potential for antigen-loss variants as an escape mechanism. Their data does little to assuage that possibility for some tumors, but clearly others do not demonstrate the phenomenon. The authors have presented thorough and forthright responses to the issues raised in the initial review and I have no further questions.

Thanks for the positive remarks about the manuscript. Further pre-clinical work will aim to elucidate resistance mechanisms to this therapy and design novel therapies to address them. As the reviewer remarks, we have performed in-depth studies in mouse models and not found evidence of known toxicities associated with targeting melanosomal proteins. We will monitor potential ocular, hearing and skin toxicities closely in the clinical trial and mitigate them with local treatment if possible or systemic treatment to stop CAR-T cell activity if required.

Reviewer #3 (Remarks to the Author):

I believe the conclusions of the manuscript are well supported and the authors have answered all of the questions raised by the original reviewers adequately.

I have only one minor comment that I personally found the title on line 281 confusing as I was not sure what panel was being referred to.

We thank the reviewer for the positive comments and are glad they are satisfied with our answers to the original revision comments. We have changed the title on line 281 to clarify its meaning.